# Covariance estimation using Markov chain Monte Carlo

**Yunbum Kook** [1]   **Matthew S. Zhang** [2][3]

## Abstract

We investigate the complexity of covariance matrix estimation for Gibbs distributions based on dependent samples from a Markov chain. We show that when the distribution satisfies a Poincaré inequality and the chain possesses a spectral gap, we can achieve similar sample complexity using MCMC as compared to an estimator constructed using i.i.d. samples, with potentially much better query complexity. As an application of our methods, we show improvements for the query complexity in both constrained and unconstrained settings for concrete instances of MCMC. In particular, we present streamlined and principled analysis of isotropic rounding procedures for sampling uniformly on convex bodies.

## 1. Introduction

High-dimensional sampling and convex body rounding are cornerstones of modern statistics and scientific computing. A fundamental ingredient within both procedures is accurate estimation of the covariance matrix of a target distribution. In the convex body setting, rounding algorithms use this matrix to place the body in (approximate) isotropic position so that its covariance is close to identity, which then accelerates various computational procedures such as volume estimation, sampling, and optimization (Kannan et al., 1997; Lovász & Vempala, 2006; Jia et al., 2021). Similarly, in Bayesian inference, fast covariance estimation allows for efficient quantification of the posterior uncertainty from Markov chain Monte Carlo outputs (Neal et al., 2011; Girolami & Calderhead, 2011).

In this work, we focus on the problem of mean and covariance estimation using Markov chain Monte Carlo (MCMC)

[1]School of Computer Science, Georgia Institute of Technology, Atlanta, Georgia, USA [2]Department of Computer Science, University of Toronto, Toronto, Ontario, Canada [3]Vector Institute, Toronto, Ontario, Canada. Correspondence to: Yunbum Kook <yb.kook@gatech.edu>, Matthew S. Zhang <matthew.zhang@mail.utoronto.ca>.

*Proceedings of the 43$^{rd}$ International Conference on Machine Learning*, Seoul, South Korea. PMLR 306, 2026. Copyright 2026 by the author(s).

methods. One setting which often arises in practice is when we want to approximately estimate these statistics for a distribution $\pi \propto \exp(-V)$, where we are only given query access to $V : \mathbb{R}^d \to \mathbb{R}$ and its gradient. As it is intractable in general to obtain the normalizing constant for the measure, one can only hope to obtain approximate samples from $\pi$ via a MCMC procedure.

Precisely, suppose we are given some $\pi$-stationary Markov kernel $P : \mathbb{R}^d \times \mathcal{B}(\mathbb{R}^d) \to \mathbb{R}$, with $P(A \mid x) \coloneqq P(x, A)$ and an initial tractable probability measure $\pi_0$. Then, the MCMC scheme corresponding to $(\pi_0, P)$ generates the sequence $\{X_k\}_{k \in [N]}$ through the procedure

$$X_0 \sim \pi_0 \,, X_k \sim P(\cdot \mid X_{k-1}) \text{ for all } k = 1, \ldots, N \,. \quad (1.1)$$

The cost for this algorithm is quantified by the total number $N_{\text{tot}}$ of queries; we are interested in its dependence on $d$ and other problem parameters to be defined later. Throughout the paper, *sample complexity* refers to the number of samples used in the estimator, whereas *query complexity* refers to the total number of oracle calls needed to generate those states. The main results in §3 are sample-complexity statements; query complexity is evaluated only after specifying the sampler and oracle model in the applications.

The general computational theory for covariance estimation assumes *independent and identically distributed* (i.i.d.) samples. However, in practice, statisticians take them from a sequence generated by some Markov chain; they use the iterates $\{X_{N_{\text{burn}}+kN_{\text{mix}}}\}_{k \le N_{\text{samp}}}$ to construct their estimates, with the "burn-in" time $N_{\text{burn}}$ needed to generate a good initial sample. From this, the next $N_{\text{samp}}$ samples are collected every $N_{\text{mix}}$ steps and used for the statistical estimation process, with $N_{\text{mix}}$ chosen so that samples are (approximately) uncorrelated and $N_{\text{samp}}$ chosen so that we can construct an estimator that is suitably concentrated around the quantity of interest. Given that in most circumstances the cost of generating a *warm*-start (i.e., finding a $\pi_0$ that is $O(1)$-close to $\pi$) dominates that of bringing $\pi_0$ to $\pi$ within desired accuracy (i.e., $N_{\text{burn}} \gg N_{\text{mix}}$), it is much more cost-effective to continue evolving the Markov chain from the previous sample, rather than restarting from ground up. This is also the workflow used in many applied MCMC pipelines: one long chain is run after burn-in, thinned or subsampled, and then used to estimate posterior means and covariances. Our results give a non-asymptotic justification for this proce-

dure under verifiable functional-inequality and spectral-gap assumptions.

To illustrate, when attempting to find the covariance matrix of the uniform measure on a well-rounded convex body, achieving constant-factor control requires $\widetilde{O}(d^4)$ queries when using independent samples. Within unconstrained sampling, if the target distribution is sufficiently regular, we would expect to incur $d^{5/2}$-dependence in query complexity if our starting measure is not warm. On the other hand, it is known that in both instances, sampling from a warm-start incurs fewer queries by a factor of $d$. Thus, determining the sample and query complexity needed for accurate statistical estimation with dependent samples is of paramount importance.

However, naïvely applying the standard i.i.d. concentration analysis for the resulting estimands constructed with *dependent* samples does not lead to a useful bound. Indeed, without leveraging an additional structural property for $P$ such as its spectral gap, the estimand could never concentrate even as $N_{\text{samp}} \to \infty$ (for instance when $P = \text{id}$). Instead, the present work operates in the setting where $\pi$ satisfies a **Poincaré inequality** (PI) with constant $C_{\text{PI}}$, and also assumes that $P$ possesses a **spectral gap** of $\lambda > 0$. See §2 for definitions of these two concepts; intuitively, the Poincaré inequality implies that the tail of $\pi$ should decay on the order of $\exp(-\|x\|/C_{\text{PI}}^{1/2})$, while the spectral gap guarantees that law $X_k$ converges exponentially to $\pi$ in an $L^2$ sense.

Encouragingly, Neeman et al. (2024) shows that under suitable assumptions on spectral gap, the sum of a Markovian sequence of matrices will satisfy a Bernstein-type inequality, and thereby provides sample complexity bounds for the estimation step. On the other hand, (PI) typically implies rapid mixing of the Markov chain, which controls the query complexity of the algorithm. Motivated by these efforts, we investigate the following research question:

**Question:** What is the sample and query complexity of mean and covariance estimation for a probability measure $\pi$ which satisfies a Poincaré inequality (PI)? Furthermore, what does it imply for various practical applications?

## 1.1. Results

**Covariance estimation via MCMC.** We demonstrate that MCMC methods for distributions satisfying (PI) reduce the query complexity by a significant factor, compared to an estimator constructed using i.i.d. samples. We summarize our main result in the theorem below. A similar guarantee also holds for the mean $\overline{X} := N^{-1} \sum_{i=1}^N X_i \approx \mathbb{E}_\pi X$.

**Theorem 1.1** (Informal; see Theorem 3.4)**.** *Assume that $\pi$ satisfies a Poincaré inequality and $P$ has a spectral gap. For $\varepsilon > 0$ and $\delta \in (0, d)$, the estimator $\overline{\Sigma} :=$*

$\frac{1}{N} \sum_{i=1}^N (X_i - \overline{X})^{\otimes 2}$, *where $u^{\otimes 2} = uu^\mathsf{T}$, satisfies with high probability that $|\overline{\Sigma} - \text{cov}\,\pi| \preceq \varepsilon\Sigma + \delta I_d$, so long as $N = \widetilde{O}(\frac{\text{tr}\,\Sigma + C_{\text{PI}}(\pi)}{\varepsilon\delta})$.*

See its multiplicative form in Corollary 3.6. To the best of our knowledge, we are the first work to give a theory of covariance estimation using MCMC under the standard assumptions of a Poincaré inequality and a spectral gap.

**Applications.** Our framework leads to a streamlined statistical analysis, which addresses sample dependence through a principled approach based on the spectral gap and Poincaré constant. This enables us to avoid the convoluted and ad-hoc analyses used in prior work. To illustrate this, we provide domains where our framework can be leveraged for streamlined analyses and improvements in query complexity.

The first occurs when sampling uniformly from a convex body $\mathcal{K} \subset \mathbb{R}^d$ (i.e., $\pi \propto \mathbb{1}_\mathcal{K}$). This problem initiated the quantitative study of covariance estimation in theoretical computer science (see §1.2). We demonstrate that, when $\mathcal{K}$ is well-rounded, $\mathbb{E}_\pi[\|\cdot\|^2] \lesssim d$, we can estimate the covariance up to a constant factor with probability $1 - O(1/d)$, using roughly $d^3$ oracle queries in expectation (Lemma 4.3). This contrasts with the best known complexity of roughly $d^4$ in the setting using i.i.d. samples. Using this, we can therefore convert a well-rounded convex body to a near-isotropic one with the same complexity (Lemma 4.6). This is a key ingredient in the *isotropic rounding* procedure of Jia et al. (2021), needed for the state-of-the-art complexity in uniform constrained sampling. Alongside this, we provide a simpler and more rigorous proof of the rounding algorithm therein (see §4.1). We remark that Kook & Vempala (2025) relies on our framework and proof when generalizing the isotropic rounding scheme to general log-concave distributions.

Secondly, we show that for unconstrained sampling $\pi \propto \exp(-V)$, when $\pi$ satisfies (PI) with constant $\alpha^{-1}$ and $V$ is $\beta$-smooth, we achieve a covariance estimate $\|\overline{\Sigma} - \Sigma\| \leq \varepsilon\|\Sigma\|$ with query complexity $\widetilde{O}(\kappa d^{3/2}/\varepsilon^2)$ for $\kappa := \beta/\alpha$. If instead one were to use i.i.d. samples, then the best known complexity is $O(\kappa d^{5/2}/\varepsilon^2)$, which is worse by a dimensional factor. See Theorem 4.14.

## 1.2. Related work

**Covariance estimation.** Covariance estimation has long been studied in statistics. We concentrate on the following setting: *given random i.i.d. samples $(X_i)_{i \leq N}$ from a centered distribution $\pi$ with $\Sigma := \text{cov}\,\pi$, bound the number $N$ of samples for which $\|\frac{1}{N} \sum_{i=1}^N X_i^{\otimes 2} - \Sigma\| \leq \frac{1}{10}\|\Sigma\|$ holds with high probability*. In this framework, the problem was first addressed by Kannan et al. (1997), obtaining $N = O(d^2)$ for a uniform distribution over a convex body. Their analysis occurred in the context of isotropic round-

ing of a convex body, which was one of the subroutines in their volume-computation algorithm. This was improved for general distributions by Bourgain (1996); Rudelson (1999), and by Paouris (2006) for a convex body. The main technique used by all of these works was a noncommutative moment inequality. Then, the seminal work of Adamczak et al. (2010) showed via a chaining argument that $N = O(d)$ for any log-concave distribution. They also showed that this holds for a sub-exponential distribution if additionally $\|X\| \lesssim d^{1/2}$ (see Remark 4.11 therein). This bound *without logarithmic overhead* was further extended by Srivastava & Vershynin (2013) under a bounded moment condition.

Another line of work follows from the matrix Laplace transform method, pioneered by Ahlswede & Winter (2002) and further developed in Oliveira (2009); Vershynin (2012); Tropp (2012). The eventual bound obtained is also $N = O(d \log d)$, where logarithmic oversampling is in general unavoidable. For a more comprehensive survey of the literature, see the following monographs: Vershynin (2012; 2018); Rigollet & Hütter (2023).

**Concentration inequalities under dependence.** Concentration inequalities such as the matrix Bernstein inequality serve as a general arsenal for establishing the sample complexity of this type of problem in the setting of *i.i.d. data*. We refer readers to Tropp (2015) for a comprehensive overview of this topic. By contrast, the setting where $\{X_1, \ldots, X_n\}$ are correlated has been comparatively understudied. Beginning with Gillman (1993), several works (Lezaud, 1998; Paulin, 2015; Jiang et al., 2018; Fan et al., 2021) have proven concentration inequalities for (scalar) functions of iterates arising from a Markov process.

Matrix concentration results under Markovian assumptions have only recently been studied (Garg et al., 2018; Qiu et al., 2020). Most relevant to our work is that of Neeman et al. (2024), which establishes a Bernstein inequality in the Frobenius norm, assuming that each sample is bounded and that the Markov chain has a spectral gap. We also note that other literature has shown concentration for correlated iterates under conditions other than a spectral gap (Mackey et al., 2014; Paulin et al., 2016). In particular, Bernstein inequalities have been established under other assumptions such as the $\beta$-mixing or $\tau$-mixing of the chain. Under these properties, the representative works (Merlevedé et al., 2009; Merlevède et al., 2011; Banna et al., 2016) show a Bernstein-like matrix concentration inequality, analogous to Neeman et al. (2024). However, in our applications it is difficult to verify these conditions, unlike the spectral gap condition considered in our work.

Although matrix Bernstein inequalities are indeed very strong, they need to be combined with a tail estimate in order to be usable in our setting. Adamczak et al. (2010)

shows that this is possible under a log-concavity assumption of the target distribution. However, their argument cannot be applied out-of-the-box to Markovian iterates, and requires additional adjustments to remain valid.

**Sampling under functional inequalities.** The relationship between Markov chain methods and spectral gaps dates back to the early theory of MCMC (Mihail, 1989; Fill, 1991), and has been thoroughly characterized in discrete space.

In continuous state-space, however, the problem is complicated by the need to ensure implementability of the resulting algorithm. Early results characterized the convergence under log-concavity or log-Sobolev conditions (Dalalyan & Tsybakov, 2012; Dalalyan, 2017; Durmus et al., 2019; Vempala & Wibisono, 2019). While the continuous-time convergence of many idealized processes can be derived easily from Poincaré-type inequalities (Bakry et al., 2014), only recently were results proven under (PI) for implementable samplers such as the Langevin Monte Carlo algorithm or the proximal sampler (Lehec, 2023; Chewi et al., 2022; Chen et al., 2022; Altschuler & Chewi, 2024). These works demonstrate that when $\pi$ satisfies (PI), the proximal sampler actually contracts in $\chi^2$-divergence for any input measure. This is equivalent to a spectral gap for the proximal sampler. As a result, we focus our attention on the proximal sampler, since merely guaranteeing $\varepsilon$-accuracy in $\chi^2$-divergence is not sufficient for an estimand constructed from Markovian iterates to concentrate.

For constrained targets such as uniform distributions over convex bodies, Ball walk (Lovász & Simonovits, 1993; Kannan et al., 1997) and Hit-and-run (Smith, 1984; Lovász, 1999) were shown to have mixing times depending on the Cheeger isoperimetric constant of the target. Only recently, Kook et al. (2024); Kook & Zhang (2025); Kook et al. (2026) proposed a sampler with Rényi guarantees which directly quantify the dependence on the Poincaré constant; this in turn depends on the degree of *isotropy* of the target, i.e., how close its covariance matrix is to the identity. An extension of their approach to general log-concave distributions was achieved in Kook & Vempala (2025).

A related line of research (Lovász & Vempala, 2006; Cousins & Vempala, 2018; Jia et al., 2021) discovered a procedure which could place any convex body in isotropic position (and this was extended to general log-concave distributions by Kook & Vempala (2025)). They iteratively estimate the covariance matrix of subsets of the convex body, and then sample from it with only $\widetilde{O}(d^{3.5})$ complexity. To achieve their complexity, they rely on a Markovian covariance estimator subroutine, which we also study in the present work.

**Organization.** In §2, we provide some preliminaries needed for our main result. In §3, we state our main bound on the concentration of covariance matrices generated by a Markov chain, and present a brief proof sketch. §4 then gives our two primary applications for covariance estimation, and includes some additional details for the isotropic rounding algorithm, with all the proofs in §3 and §4 deferred to §A and §B. We conclude by summarizing future directions along this line of research.

## 2. Preliminaries

We use $\|\cdot\|$ for the operator norm on $\mathbb{R}^{d \times d}$. For a vector $u \in \mathbb{R}^d$, we denote its outer product by $u^{\otimes 2} := uu^\mathsf{T}$. Next, $a = O(b)$ or $a \lesssim b$ signifies that $a \leq cb$ for an absolute constant $c > 0$. Similarly, $a \gtrsim b, a = \Omega(b)$ signifies $a \geq cb$, while $a = \Theta(b)$ signifies $a \lesssim b, b \lesssim a$ simultaneously. We also use $a = \widetilde{O}(b)$ to denote $a = O(b)$ up to a polylogarithmic factor. For matrices $A, B$, $|A| \preceq B$ indicates $-B \preceq A \preceq B$. We conflate a measure and its density when there is no confusion. $\mathcal{P}(\mathbb{R}^d)$ indicates the set of probability measures over $\mathbb{R}^d$.

Since we are primarily concerned with the high-dimensional complexity of the algorithm, many of our results involve algebraic simplifications that are valid only when the dimension $d$ is sufficiently large. In quantifying failure probabilities for our algorithm, we introduce a constant $\mathfrak{p}$ which may change from line to line, but is always some universal constant that does not depend on $d, \varepsilon$ or any other parameter of interest. We now formalize key notions needed for our main results.

**Definition 2.1.** We say that a probability measure $\pi$ on $\mathbb{R}^d$ satisfies a *Poincaré inequality* (PI) with parameter $C_{\mathsf{PI}}(\pi) \geq 0$ if for all smooth functions $f : \mathbb{R}^d \to \mathbb{R}$,

$$\mathrm{var}_\pi f \leq C_{\mathsf{PI}}(\pi) \, \mathbb{E}_\pi[\|\nabla f\|^2] \,, \qquad \text{(PI)}$$

where $\mathrm{var}_\pi f := \mathbb{E}_\pi[|f - \mathbb{E}_\pi f|^2]$. Here "smooth" may be read as continuously differentiable with square-integrable gradient; the $\beta$-smoothness assumption in §4 refers instead to Lipschitz continuity of $\nabla V$.

(PI) implies that any 1-Lipschitz function $f$ concentrates sub-exponentially around its mean $\mathbb{E}_\pi f$:

$$\mathbb{P}_\pi(f - \mathbb{E}_\pi f \geq t) \leq 3 \exp\big(-t/C_{\mathsf{PI}}^{1/2}(\pi)\big) \quad t \geq 0 \,. \quad (2.1)$$

**Definition 2.2.** Let $P : \mathbb{R}^d \times \mathcal{B}(\mathbb{R}^d) \to \mathbb{R}$ be a Markov kernel, i.e., $P(\cdot \mid x) \in \mathcal{P}(\mathbb{R}^d)$ for every $x \in \mathbb{R}^d$. For the stationary measure $\pi \in \mathcal{P}(\mathbb{R}^d)$ of the Markov kernel, $P$ is said to have a *spectral gap* $\lambda \in (0, 1]$ if for every $f : \mathbb{R}^d \to \mathbb{R}$ with $f \in L^2(\pi)$ and $\mathbb{E}_\pi f = 0$, we have $\|Pf\|_{L^2(\pi)}^2 := \mathbb{E}_{X \sim \pi}\big[\big(\mathbb{E}_{Y \sim P(\cdot|X)}[f(Y)]\big)^2\big] \leq (1-\lambda)^2 \|f\|_{L^2(\pi)}^2$. A kernel $P$ is *reversible* if $\langle Pf, g \rangle_{L^2(\pi)} = \langle f, Pg \rangle_{L^2(\pi)}$ for all $f, g \in L^2(\pi)$.

We say a sequence $(X_0, X_1, \dots)$ is *driven by* $P$ with initial distribution $\pi_0$ if it is generated by the procedure in (1.1). The following metrics[1] between probability measures will be useful throughout our work, particularly for the applications.

**Definition 2.3** (Probability divergences). For $\mu, \nu \in \mathcal{P}(\mathbb{R}^d)$, the $\chi^q$-*divergence* is defined by

$$\chi^q(\mu\|\nu) := \int \big(\frac{\mathrm{d}\mu}{\mathrm{d}\nu}\big)^q \mathrm{d}\nu - 1 \quad \text{if } \mu \ll \nu, \text{ and } \infty \text{ otherwise.}$$

In particular, $\chi^2(\mu \| \nu) = \int (\frac{\mathrm{d}\mu}{\mathrm{d}\nu} - 1)^2 \, \mathrm{d}\nu$. The *q-Rényi divergence* and *total variation* distance between the two measures is given by

$$\mathcal{R}_q(\mu \| \nu) := \tfrac{1}{q-1} \log\big(\chi^q(\mu \| \nu) + 1\big) \,,$$
$$\|\mu - \nu\|_{\mathsf{TV}} := \sup_{B \in \mathcal{F}} |\mu(B) - \nu(B)| \,,$$

where $\mathcal{F}$ is the set of all measurable subsets of $\mathbb{R}^d$. We also use the standard convention $\mathcal{R}_\infty(\mu \| \nu) = \log \mathrm{ess} \sup \frac{\mathrm{d}\mu}{\mathrm{d}\nu}$.

## 3. Covariance estimation using dependent samples

**Assumption 3.1.** $P$ is a reversible Markov kernel on $\mathbb{R}^d$ with stationary distribution $\pi$ and spectral gap $\lambda$ (Definition 2.2). Let $(X_1, \dots, X_N)$ be a sequence driven by $P$ with initial distribution $\pi_0 = \pi$.

The choice of $\pi_0 = \pi$ can be removed at the cost of a multiplicative factor $\|\frac{\mathrm{d}\pi_0}{\mathrm{d}\pi}\|_\infty$ in our failure probabilities (Lemma A.1). The following is instrumental in establishing covariance bounds.

**Theorem 3.2** ((Neeman et al., 2024), Theorem 2.2). *Under Assumption 3.1, suppose that $F_i : \mathbb{R}^d \to \mathbb{R}^{d \times d}$,[2] for $i \in [N]$ is a sequence of functions each mapping to real symmetric $d \times d$ matrix satisfying $\mathbb{E}_\pi[F_i(\cdot)] = 0$, $\|\mathbb{E}_\pi[F_i^2(\cdot)]\| \leq \mathcal{V}_i$, and $\sup_{\mathbb{R}^d} \|F_i(\cdot)\| \leq \mathcal{M}$. For $\sigma^2 := \sum_{i=1}^N \mathcal{V}_i$, it holds that*

$$\mathbb{P}\Big(\Big\|\sum_{i=1}^N F_i(X_i)\Big\| \geq t\Big) \leq d^{2 - \frac{\pi}{4}} \exp\big(-\tfrac{t^2/(32/\pi^2)}{\alpha(\lambda)\,\sigma^2 + \beta(\lambda)\,\mathcal{M}t}\big) \,,$$

*where $\|\cdot\|$ is the operator norm, and $\alpha(\lambda) = \frac{2-\lambda}{\lambda}$ and $\beta(\lambda) = \frac{8/\pi}{\lambda}$.*

When $\lambda$ is some absolute constant bounded away from $0$, the quantity in the exponential is $\mathcal{O}(\min(t^2/\sigma^2, t/\mathcal{M}))$, i.e., quadratic for small $t$ and linear for large $t$.

As a corollary, we can deduce similar guarantees for vectors/non-square matrices (see §A.1).

---

[1] Although not all of these are proper metrics, we still refer to them as such in sense of "performance metric".

[2] This is valid even for arbitrary input state spaces, but this will not concern us in this work.

**Corollary 3.3.** *Under Assumption 3.1, suppose that* $\mathbb{E}_\pi X_i = 0$, $\mathbb{E}_\pi[\|X_i\|^2] \leq \mathcal{V}_i$, *and* $\|X_i\| \leq \mathcal{M}$ *almost surely for* $i \in [N]$. *Then, denoting* $\sigma^2 = \sum_{i=1}^N \mathcal{V}_i$ *and where* $\alpha, \beta$ *are as in Theorem 3.2, the following holds for all* $t \geq 0$,

$$\mathbb{P}\Big(\Big\|\sum_{i=1}^N X_i\Big\| \geq t\Big) \leq (d+1)2^{-\frac{\pi}{4}}\exp\Big(-\frac{t^2/(32/\pi^2)}{\alpha(\lambda)\,\sigma^2 + \beta(\lambda)\,\mathcal{M}t}\Big).$$

### 3.1. Main results

Suppose our Markov chain is nice enough that the spectral gap is $\lambda \geq 0.99$, which can always be done by composing a sufficient number of iterations of a more tractable chain with smaller spectral gap. Then our estimators for the mean and covariance are respectively

$$\overline{X} := \frac{1}{N}\sum_{i=1}^N X_i\,,$$

$$\overline{\Sigma} := \frac{1}{N}\sum_{i=1}^N (X_i - \overline{X})^{\otimes 2} = \frac{1}{N}\sum_{i=1}^N X_i^{\otimes 2} - \overline{X}^{\otimes 2}\,.$$

We state our covariance estimation results in both additive and multiplicative forms.

**Theorem 3.4** (Additive form). *Under Assumption 3.1, if* $\pi$ *also satisfies* (PI) *and* $P$ *has a spectral gap* $\lambda \geq 0.99$, *the covariance estimator* $\overline{\Sigma} = \frac{1}{N}\sum_{i=1}^N (X_i - \overline{X})^{\otimes 2}$ *satisfies that for any* $\varepsilon > 0$ *and* $\delta \in (0, d)$, *with probability at least* $1 - \mathfrak{p}/d$, $|\overline{\Sigma} - \Sigma| \preceq \varepsilon\Sigma + \delta I_d$, *so long as* $N \succsim \frac{\text{tr}\,\Sigma + C_{\text{PI}}(\pi)}{\varepsilon\delta}\log^2 d\,\log^2 \frac{d\max(d, C_{\text{PI}}(\pi) + \text{tr}\,\Sigma)}{\delta}$.

**Theorem 3.5** (Multiplicative form). *In the setting of Theorem 3.4,* $\overline{\Sigma}$ *satisfies that for any* $\varepsilon \in (0, 1)$, *with probability at least* $1 - \mathfrak{p}/d$, $\|\overline{\Sigma} - \Sigma\| \leq \varepsilon\|\Sigma\|$, *if* $N \succsim \frac{d}{\varepsilon^2}\frac{\text{tr}\,\Sigma + C_{\text{PI}}(\pi)}{\text{tr}\,\Sigma}\log^2 d\,\log^2 \frac{d(C_{\text{PI}}(\pi) + \text{tr}\,\Sigma)}{\varepsilon\|\Sigma\|}$.

**Corollary 3.6** (Multiplicative form; spectral). *In the setting of Theorem 3.4,* $\overline{\Sigma}$ *satisfies for any* $\varepsilon \in (0, 1)$, *with probability at least* $1 - \mathfrak{p}/d$, $|\overline{\Sigma} - \Sigma| \preceq \varepsilon\Sigma$, *if* $N \succsim \frac{d + C_{\text{PI}}(\nu)}{\varepsilon^2}\log^2 d\,\log^2 \frac{d(d + C_{\text{PI}}(\nu))}{\varepsilon}$, *where* $\nu := (\Sigma^{-1/2})_{\#}\pi$.

*Remark* 3.7 (Poincaré constant of $\nu$). For $\nu = (\Sigma^{-1/2})_{\#}\pi$, it holds in general that $C_{\text{PI}}(\nu) \leq \lambda_1^{-2}C_{\text{PI}}(\pi)$ for the smallest eigenvalue $\lambda_1$ of $\Sigma$. In particular, it is well-known by Klartag (2023) that a log-concave distribution $\pi$ satisfies $\|\Sigma\| \leq C_{\text{PI}}(\pi) \lesssim \|\Sigma\|\log d$.

*Remark* 3.8 (Independent samples). As a corollary, we can obtain an analogous result when the samples are independent by considering the case where the kernel $P(\cdot\,|\,x) = \pi$ identically. In this case, the spectral gap is 1, and the previous results can apply without alteration. As far as we can tell, this is the first time that such a result has been explicitly presented for distributions satisfying (PI), which may be of independent interest.

**Proof sketch.** We sketch a proof for Theorem 3.4 below, deferring the detailed analysis to §A. In our analysis, we show that the error can be divided into two terms

$$\overline{\Sigma} - \Sigma = \frac{1}{N}\sum_{i=1}^N\big((X_i - \mu)^{\otimes 2} - \Sigma\big) - (\overline{X} - \mu)^{\otimes 2}\,.$$

The techniques we use for the mean and covariance error terms will be similar, so we shall concentrate on the argument for the first term. We split this into three sub-terms, similar to the argument in Adamczak et al. (2010), where $B \subseteq \mathbb{R}^{N \times d}$ will be some suitably "nice" set,

$$\frac{1}{N}\sum X_i^{\otimes 2} - \Sigma = \frac{1}{N}\sum_{i=1}^N (X_i^{\otimes 2}\mathbb{1}_B - \mathbb{E}[X_i^{\otimes 2}\mathbb{1}_B])$$
$$+ \frac{1}{N}\sum_{i=1}^N X_i^{\otimes 2}\mathbb{1}_{B^c} - \mathbb{E}[X^{\otimes 2}\mathbb{1}_{B^c}]\,,$$

denoting the three terms on the right side as $\mathsf{A}_1, \mathsf{A}_2, \mathsf{A}_3$ respectively. Since the first term relates to the concentration of a bounded matrix around its mean, it can be handled using Theorem 3.2 after some detailed calculations (Lemma A.7). As for the rest, we would like to use a concentration inequality under (PI) to ensure that $\mathsf{A}_2, \mathsf{A}_3$ are small. While the argument of Adamczak et al. (2010) uses independence of the iterates to establish an exponential tail decay of $\|X\|$, our Markovian argument uses the sub-exponential one-sample tail from the Poincaré inequality but controls the empirical tail average by Markov's inequality, yielding only polynomial decay in the failure probability for this part. This is sufficient for Theorem 3.4. Nonetheless, we also show a slightly improved tail bound in §C.

## 4. Applications

### 4.1. Isotropic rounding via uniform sampling

As a first application, we consider a version of the "sampling-and-rounding" scheme used in Jia et al. (2021), and simplify analysis of their algorithm. This is summarized in the following problem, which is a keystone in rounding schemes for general convex bodies (see Remark B.2). A membership oracle for $\mathcal{K}$ answers queries of the form "is $x \in \mathcal{K}$?"; query counts in this subsection are in expectation because the sampler uses rejection steps with a random number of oracle calls.

**Problem 4.1.** Let $\mathcal{K} \subset \mathbb{R}^d$ be a well-rounded convex body containing a ball of radius 1, meaning that its uniform distribution $\pi_{\mathcal{K}}$ satisfies $\mathbb{E}_{\pi_{\mathcal{K}}}[\|\cdot\|^2] \lesssim d$. Given membership-oracle access to $\mathcal{K}$, construct an affine map $A(x) = M(x - \widehat{\mu})$ such that the pushforward $A_{\#}\pi_{\mathcal{K}}$ is $c$-isotropic for a universal constant $c \approx 1$, i.e., $c^{-1}I_d \preceq \text{cov}(A_{\#}\pi_{\mathcal{K}}) \preceq cI_d$, using $\widetilde{O}(d^3)$ expected membership queries.

We note that if we obtain the mean $\mu$ and covariance $\Sigma$ of $\pi$, then the transformed convex body $\Sigma^{-1/2}(\mathcal{K} - \mu)$ is isotropic (i.e., $\mathbb{E}X = 0$ and $\mathbb{E}[X^{\otimes 2}] = I_d$); this procedure is termed *rounding*. If one insists on generating i.i.d. samples from scratch, then a single approximately uniform sample costs about $d^3$ queries and about $d$ samples are needed for covariance estimation, leading to a $d^4$ cost. The saving comes from paying the warm-start cost once and then taking dependent samples from the same chain. The algorithm of Jia et al. (2021) implements this idea through repeated sampling, approximate covariance estimation, and rescaling; we compare their dependence argument with ours in §4.1.2.

### 4.1.1. ALGORITHM

**Uniform sampling.** Kook et al. (2024) proposes In-and-Out for uniformly sampling from $\mathcal{K}$ containing a ball of radius $r$. It iterates two steps with $h \asymp r^2/d^2$: (i) $y_{i+1} \sim \mathcal{N}(x_i, hI_d)$ and (ii) $x_{i+1} \sim \mathcal{N}(y_{i+1}, hI_d)|_{\mathcal{K}}$, where (ii) is implemented by rejection sampling with proposal $\mathcal{N}(y_{i+1}, hI_d)$. From a warm start, In-and-Out iterates $n \lesssim qh^{-1}C_{\mathsf{PI}}(\pi)$ polylog $\frac{1}{\delta\varepsilon}$ times to find a sample $X_n$ with $\mathcal{R}_q(\text{law } X_n \parallel \pi) \leq \varepsilon$, using $\widetilde{O}(n)$ queries for success probability $1 - \delta$. Denoting by $P$ the Markov kernel of one iteration, In-and-Out satisfies an exponential contraction in $\chi^2$,

$$\chi^2(\mu P \parallel \pi) \leq \frac{\chi^2(\mu \parallel \pi)}{(1 + h/C_{\mathsf{PI}}(\pi))^2}. \qquad (4.1)$$

The spectral gap of $P$ is roughly lower bounded by $hC_{\mathsf{PI}}^{-1}$. Thus, we can ensure that the spectral gap of a new Markov kernel $P^n$ for $n \lesssim h^{-1}C_{\mathsf{PI}}$ is at least 0.99. We denote In-and-Out$_N(\mu, \nu, h)$ for the Markov chain with kernel $P^N$, initial distribution $\nu$, and target distribution $\mu$.

**High-level description.** Let $\pi_{\mathcal{K}}$ denote the uniform distribution over a convex body $\mathcal{K}$, and $r := \mathsf{inrad}(\mathcal{K})$ denote the radius of the largest ball contained in $\mathcal{K}$. Recall that In-and-Out needs roughly $d^2\|\Sigma\|/r^2$ queries per sample from a warm start, which indicates that the mixing of the sampler suffers from the skewness of $\mathcal{K}$ (i.e., how far $\Sigma$ is from $I_d$).

Algorithm 1, a modified version of Jia et al. (2021, Algorithm 2), starts by generating a warm point via Gaussian cooling (Kook & Zhang, 2025) using $d^3$ membership queries. At iteration $i$, the certified inradius is $r_i$, and a warm-started In-and-Out sample costs roughly $d^2\|\Sigma_i\|/r_i^2$ queries. The algorithm draws $k_i = \widetilde{O}(C^2r_i^2)$ retained samples from one chain to obtain a covariance estimate. It then computes the projection $P_i$ onto the eigenspaces of $\widehat{\Sigma}_i$ with eigenvalue at most $d$ and applies $M_i = I_d + P_i$, thereby doubling precisely those directions. This almost doubles the certified inradius. Although $k_i$ increases by about a factor of four after $r_i$ doubles, this cancels the $r_i^{-2}$ factor in the

---

**Algorithm 1** Isotropize – See Algorithm 3 for more details.

**Input:** $\mathcal{K} \subset \mathbb{R}^d$ and $T_1 \in \mathbb{R}^{d \times d}$ such that $\mathcal{K}_1 := T_1\mathcal{K}$ satisfies $\frac{1}{4} \leq \mathsf{inrad}(\mathcal{K}_1)$ and $\mathbb{E}_{\pi_1}[\|\cdot\|^2] \lesssim d$.
**Output:** the affine map $A(x) = \widehat{\Sigma}^{-1/2}(T_ix - \widehat{\mu})$.

1: Let $\pi_i$ denote $\pi_{\mathcal{K}_i}$, and $\Sigma_i$ be its covariance.
2: Run Gaussian cooling to generate an $O(1)$-warm point $Z_1$ for $\pi_1$. Let $r_1 = 1/4$ and $i = 1$.
3: **while** $r_i^2 \lesssim d$ **do**
4:     Draw $\{X_{i,j}\}_{j\in[k_i]} \leftarrow$ In-and-Out$_{N_i}\left(\pi_i, \delta_{Z_i}, \frac{r_i^2}{d^2}\right)$ with $N_i = \frac{d^3}{r_i^2}$ and $k_i = r_i^2$.
5:     For $\widehat{\mu}_i = \frac{1}{k_i}\sum_{j=1}^{k_i} X_{i,j}$ and $\widehat{\Sigma}_i = \frac{1}{k_i}\sum_{j=1}^{k_i}(X_{i,j} - \widehat{\mu}_i)^{\otimes 2}$, compute the orthogonal projection $P_i$ to the subspace spanned by eigenvectors of $\widehat{\Sigma}_i$ with eigenvalue at most $d$. Set $M_i = I_d + P_i$
6:     Set $T_{i+1} = M_iT_i$, $\mathcal{K}_{i+1} = M_i\mathcal{K}_i$, $Z_{i+1} = M_iZ_i$, $r_{i+1} = 2r_i(1 - 1/\log d)$, and $i \leftarrow i + 1$.
7: **end while**
8: Using $d$ samples from In-and-Out$_{d^2}(\pi_i, \delta_{Z_i}, d^{-1})$, compute the mean $\widehat{\mu}$ and covariance $\widehat{\Sigma}$.

---

per-sample cost. Lemma 4.4 keeps $\|\Sigma_i\| = \widetilde{O}(d)$, so the total query complexity over $O(\log d)$ outer loops is $\widetilde{O}(d^3)$.

### 4.1.2. COMPARISON BETWEEN APPROACHES

Jia et al. (2021) used Ball walk started at an initial distribution $\mu$, whose query complexity for obtaining a $\varepsilon$-close sample (to $\pi$) in TV is $O(Md^2\psi_{\mathsf{KLS}}^2 \log^{O(1)} 1/\varepsilon)$, where $M = \sup_{\mathcal{K}} \mu/\pi$ is a warmness parameter and $\psi_{\mathsf{KLS}}^{-1} = \inf_{S \subset \mathbb{R}^d} \pi^+(S)/\pi(S) \wedge \pi(S^c)$.

When drawing samples used for covariance estimation, Jia et al. generate an $O(1)$-warm sample and then initialize several *parallel* threads of Ball walk from it. The resulting samples are independent conditioned on that warm point, but not marginally independent. Their analysis controls this dependence through $\mu$-independence, also known as $\alpha$-mixing, together with an additional boosting step. We refer interested readers to their paper Jia et al. (2021, Computational Model), which references Lovász & Vempala (2006, §3.2).

Our variant avoids this extra layer by taking sequential samples from a single block chain and applying the Markovian covariance-estimation result of §3. The $\chi^2$ contraction of In-and-Out gives a direct spectral-gap guarantee for the block kernel. In addition, by incorporating Klartag's improvement of $C_{\mathsf{PI}}$ (or equivalently $\psi_{\mathsf{KLS}}^2$) from $\|\Sigma\| d^{o(1)}$ to $\|\Sigma\| \log d$, we bypass the anisotropic KLS bound developed in Jia et al. (2021).

Thus, the point of Lemma 4.6 is not to claim a new asymptotic bound for the well-rounded rounding subroutine al-

ready in Jia et al. (2021); rather, it shows that the same $\widetilde{O}(d^3)$ query count follows from a general Markovian covariance-estimation theorem with a transparent dependence analysis.

### 4.1.3. ANALYSIS

We analyze each algorithmic component under the event that all the previous subroutines succeed. We make all hidden constants $c, C$ (in Algorithm 1) explicit below. All proofs are deferred to §B.2.

**(1) Guarantees of the sampler.** First, we ensure that In-and-Out indeed satisfies Assumption 3.1, which is a consequence of the convergence rates established in earlier work (Kook et al., 2024).

**Lemma 4.2.** In-and-Out$_{N_i}(\pi_i, \cdot, \frac{r_i^2}{2^{10}d^2 \log(Ccr_i d)})$ with $N_i = 2^{10}C^2 d^3 r_i^{-2} \log(Ccd) \log^2 d$ has a spectral gap of at least $0.99$ (or any desired constant approaching $1$).

We present a version of Theorem 3.4 tailored to this application, with proof using the spectral-gap condition of In-and-Out and the fact that the warm point $Z_1$ is close to $\pi_1$ (see Line 2).

**Lemma 4.3.** *There exists a universal constant $c > 0$ such that, conditional on successful implementation of the* In-and-Out$_{N_i}$ *calls in the $i$-th loop, the estimator formed from $k_i$ retained outputs satisfies, with probability at least $1 - \mathtt{p}/d$, whenever $k_i \geq cd^{-1} \operatorname{tr} \Sigma_i \log^6(C^2 d)$,*

$$\frac{9}{10} \Sigma_i - \frac{d}{100} I_d \preceq \widehat{\Sigma}_i \preceq \frac{11}{10} \Sigma_i + \frac{d}{100} I_d \,.$$

Leveraging this, we can apply our Markovian covariance estimation machinery in the form of Corollary 3.6 to obtain a covariance concentration bound for Line 8.

**(2) Control over trace and operator norm.** The following lemma establishes quantitative control over changes of the trace and operator norm of the covariance $\Sigma_i$ at each iteration. We present a simpler proof of Jia et al. (2021, Lemma 3.1), without needing an anisotropic KLS bound.

**Lemma 4.4** (Covariance control)**.** *On the event that the covariance-estimation bound of Lemma 4.3 holds in every executed while-loop, the while-loop iterates at most $2 \log d$ times. Also, (1) there is a universal constant $c_{\mathrm{tr}}$ such that $\|\Sigma_i\|_1 = \operatorname{tr} \Sigma_i \leq c_{\mathrm{tr}} r_i^2 C^2 d$ and (2) $\|\Sigma_i\| \leq d(C^2 + 6i)$.*

Lastly, we prove that the inner radius $\operatorname{inrad}(\mathcal{K}_i)$ almost doubles every iteration, providing a rigorous proof of Jia et al. (2021, Lemma 3.2).

**Lemma 4.5** (inrad control)**.** *Assume $r_i \leq \operatorname{inrad}(\mathcal{K}_i)$, the covariance-estimation event of Lemma 4.3 holds in loop $i$, and $r_i^2 \leq d/(2^{10} \log^4 d)$. Then, under $r_{i+1} = 2(1 - 1/\log d)r_i$, we have $r_{i+1} \leq \operatorname{inrad}(\mathcal{K}_{i+1})$.*

Putting all of the aforementioned results together, we conclude that Algorithm 1 returns an affine map that nearly isotropizes the original body with high probability.

**Lemma 4.6.** *Algorithm 1 returns $(T_i, \widehat{\mu}, \widehat{\Sigma})$ such that $\widehat{\Sigma}^{-1/2}(T_i \mathcal{K} - \widehat{\mu}) = \{\widehat{\Sigma}^{-1/2}(T_i x - \widehat{\mu}) : x \in \mathcal{K}\}$ is $2$-isotropic with probability at least $1 - \mathtt{p}/\sqrt{d}$, using $\widetilde{O}(C^4 d^3)$ expected membership queries.*

### 4.2. Covariance estimation in unconstrained sampling

In a similar vein, we state and prove a guarantee for covariance estimation when the target is an unconstrained distribution satisfying a Poincaré inequality. Briefly, we note that the analysis of Proximal sampler (for which In-and-Out is the constrained equivalent) elegantly relates the mixing of the sampler in various divergences to the isoperimetric constants of the target $\pi$; in particular, there is a fundamental relationship between the Poincaré constant of $\pi$ and the mixing rate in $\chi^2$. This is why we use the proximal sampler in this application: its $\chi^2$ contraction directly converts the Poincaré constant into a spectral-gap guarantee for an iterated exact kernel. We are not aware of an analogous PI-to-spectral-gap guarantee for MALA, a generic Gibbs sampler, or other high-accuracy samplers under the same assumptions, although our covariance-estimation framework applies to any sampler for which such a spectral gap can be verified.

**The** Proximal sampler **for unconstrained distributions.** Below, we recall the Proximal sampler when sampling from an unconstrained distribution $\pi \propto \exp(-V)$. Proximal sampler iterates, for some step size $h > 0$: (Forward) $y_{i+1} \sim \mathcal{N}(x_i, hI_d)$ and (Backward) $x_{i+1} \sim Q_h(\cdot \,|\, y_{i+1})$, where

$$Q_h(\cdot \,|\, y_{i+1}) \propto \exp\left(-V(\cdot) - (2h)^{-1} \|\cdot - y_{i+1}\|^2\right).$$

Similar to In-and-Out, this procedure can be seen as Gibbs sampling. However, unlike In-and-Out, the implementation of the reverse step is not straightforward and requires some additional effort.

For their state-of-the-art result, the full methodology of Altschuler & Chewi (2024) uses a composite algorithm to (approximately) implement the backwards step. The composite algorithm consists of (1) the Metropolis adjusted Langevin algorithm (MALA), a high-accuracy sampler which performs well when given a warm start, and (2) the low-accuracy underdamped Langevin Monte Carlo (ULMC) sampler in order to generate that warm start. We invite the reader to peruse Altschuler & Chewi (2024) for additional details.

An important detail is that this only generates *approximate* samples from the distribution $Q_h$ of the backwards step, with some chosen error tolerance $\varrho$ so that the final output

is close to $Q_h$. We refer to the composition of the exact forward kernel with the inexact reverse kernel as $\hat{P}_{h,\varrho}$. The overall methodology is summarized in Algorithm 2.

---

**Algorithm 2** Covariance estimation in the unconstrained setting

---

**Input:** $\pi \propto \exp(-V) \in \mathcal{P}_2(\mathbb{R}^d)$ such that $V$ is $\beta$-smooth and $C_{\mathsf{PI}}(\pi) < \infty$, target error $\varepsilon > 0$.
**Output:** $(\hat{\mu}, \hat{\Sigma})$

1: Let $K = \frac{c_K d\phi}{\varepsilon^2} \log^2 \frac{d\phi}{\varepsilon} \log^4 d$, $h = \frac{1}{2\beta}$, $n_0 = c_{n_0}\kappa(d \vee \beta) \log(\kappa d)$, $n = c_n\kappa$, $\varrho = \frac{c_\varrho}{d(Kn+n_0)}$, where $c_K, c_{n_0}, c_n, c_\varrho$ are all positive universal constants, $\Sigma = \operatorname{cov}(\pi), \phi := \frac{\operatorname{tr}\Sigma + C_{\mathsf{PI}}(\pi)}{\operatorname{tr}\Sigma}, \kappa := \beta C_{\mathsf{PI}}(\pi)$.
2: Obtain $X_0 \sim \mathcal{N}(0, \beta^{-1}I_d)$.
3: **for** $j \in [K]$ **do**
4:     Draw $X_j \sim \delta_{X_{j-1}}\hat{P}_{h,\varrho}^n$.
5: **end for**
6: Compute the sample mean and covariance $\hat{\mu} \leftarrow \frac{1}{K}\sum_{j=1}^{K} X_j, \hat{\Sigma} \leftarrow \frac{1}{K}\sum_{j=1}^{K}(X_j - \hat{\mu})^{\otimes 2}$.

---

Finally, we note that the approximate kernels used at different steps are allowed to be different, and the user is free to choose the error tolerance at each step so that their final guarantee is suitably strong. For simplicity, we assume that the error tolerances do not differ from step to step.

**Results.** In this setting, we will primarily operate with the following standard assumptions.

**Assumption 4.7.** The target distribution $\pi \propto \exp(-V)$ satisfies (PI) and $V$ is $\beta$-smooth;

$$\|\nabla V(x) - \nabla V(y)\| \le \beta\|x - y\| \text{ for all } x, y \in \mathbb{R}^d.$$

For instance, when $\pi \propto \exp(-\sqrt{1 + \|x\|^2})$, we have by Bobkov (2003) that $C_{\mathsf{PI}}(\pi) = O(d)$. The perturbation principle of Holley & Stroock (1987) also states that for $\tilde{\pi} \propto \exp(-V + f)$, where $\|f\|_\infty < \infty$ and $V$ is everywhere finite, a Poincaré inequality continues to hold for $\tilde{\pi}$ with constant depending on $\|f\|_\infty$. In the sequel, we will denote the "condition number" by $\kappa := C_{\mathsf{PI}}(\pi)\beta$, which recovers the standard condition number $\beta/\alpha$ when $\pi$ is $\alpha$-strongly log-concave.

**Lemma 4.8** ((Chen et al., 2022), Theorem 4). *Suppose $\pi$ satisfies Assumption 4.7. Let $P_h$ be the Markov kernel corresponding to* Proximal sampler *with stationary measure $\pi \in \mathcal{P}(\mathbb{R}^d)$ and step size $h$. Then, $P_h$ satisfies, for any initial measure $\mu \in \mathcal{P}(\mathbb{R}^d)$*

$$\chi^2(\mu P_h \| \pi) \le \frac{\chi^2(\mu \| \pi)}{(1 + h/C_{\mathsf{PI}}(\pi))^2}.$$

In the sequel, we will also impose the following standard oracle model.

**Assumption 4.9.** Assume that we have access to oracles for $V, \nabla V$, and also that we have access to the proximal oracle with step size $h = 1/2\beta$ for $V$, which given a point $y \in \mathbb{R}^d$ returns $\arg\min_{x \in \mathbb{R}^d}\{V(x) + \frac{1}{2h}\|x - y\|^2\}$.

*Remark* 4.10. The proximal oracle is not strictly necessary, and can be removed by following the same techniques as Altschuler & Chewi (2024), at the possible cost of additional polylogarithms.

In this section, when we speak about expected query complexity, we are referring to the sum total of queries either to oracles for $V, \nabla V$ or to the proximal oracle given above. Hereafter we will also suppress the subscripts for $h, \varrho$ in the kernel, as these remain fixed throughout the algorithm. The guarantees from Altschuler & Chewi (2024, Theorem D.1) are summarized below.

**Lemma 4.11.** *Under Assumptions 4.7 and 4.9, in the setting of Lemma 4.8, there is an algorithm which, given an initial point $x \in \mathbb{R}^d$, finds $z \sim \delta_x\hat{P}$ with $\mathsf{KL}(\delta_x\hat{P} \| \delta_xP) \le \varrho$ for any $\varrho \in (0, 1/2]$, with expected query complexity $N = \widetilde{O}(d^{1/2}\log^3\frac{1}{\varrho})$.*

*Remark* 4.12. Due to its reliance on an inexact implementation of the proximal sampler, we do not know if the algorithm can be written in the form of a kernel possessing a spectral gap. As a result, we cannot apply Theorem 3.5 directly to this algorithm.

We can now convert this error into a total variation bound on the entire chain, as given below.

**Lemma 4.13.** *In the setting of Lemma 4.11, set $\nu = \mu_0P^{n_0}$ and $\hat{\nu} = \mu_0\hat{P}^{n_0}$ for any initialization $\mu_0$ and $n_0 \in \mathbb{N}$, where $\varrho \asymp \delta^2/(K\kappa+n_0)$ is chosen in Lemma 4.11, and $K \in \mathbb{N}$. For $n = O(\kappa)$, let $\nu_{1:K}, \hat{\nu}_{1:K}$ be the joint laws of $K$ iterates drawn from Markov chains with initial distributions $\nu, \hat{\nu}$ and kernels $P^n, \hat{P}^n$ respectively. Then, for a given $\delta \in (0, 1/2]$, we can guarantee that $\|\nu_{1:K} - \hat{\nu}_{1:K}\|_{\mathsf{TV}} \le \delta$. The expected query complexity of implementing the chain corresponding to $\hat{\nu}_{1:K}$ is*

$$N = \widetilde{O}(n_0d^{1/2}\log^3\frac{K}{\delta}) + \widetilde{O}(K\kappa d^{1/2}\log^3\frac{n_0}{\delta}).$$

We will take $\delta \asymp 1/\sqrt{d}$. Lemma 4.8 suggests that we set $n_0 = \kappa\log\chi^2(\mu_0 \| \pi)$, and so we need $\widetilde{O}(\kappa d^{1/2}\log\chi^2(\mu_0 \| \pi)\operatorname{polylog}K)$ queries to obtain a sample whose law is $O(1)$-close to $\pi$ in $\chi^2$-divergence. Under standard assumptions (Lemma B.3), $\log\chi^2(\mu_0 \| \pi) \asymp \widetilde{O}(d \vee \beta)$, so this complexity would be $\widetilde{O}(\kappa d^{1/2}(d \vee \beta))$. For the requisite $K$ *independent* samples, we repeat the whole procedure from scratch, expecting $\widetilde{O}(\kappa d^{1/2}(d \vee \beta)\operatorname{polylog}(K + n_0))$ queries per sample for an accurate covariance estimate.

In contrast, using $\widetilde{O}(\kappa d^{1/2}(d \vee \beta)\operatorname{polylog}K)$ queries to obtain a warm sample, a Markovian iterate-based sampler needs $\widetilde{O}(\kappa d^{1/2}\operatorname{polylog}(K + n_0))$ for each further sample.

**Theorem 4.14** (Covariance estimation in unconstrained sampling)**.** *In addition to Assumptions 4.7 and 4.9, assume that $V(0) - \min V \lesssim d$, $\mathbb{E}_\pi \|\cdot\| = \operatorname{poly}(\kappa, d)$, and $\nabla V(0) = 0$. We then have with probability $1 - \wp/\sqrt{d}$ that the matrix obtained by Algorithm 2 satisfies $\|\hat{\Sigma} - \Sigma\|_{\mathsf{op}} \leq \varepsilon \|\Sigma\|_{\mathsf{op}}$. In particular, the total query complexity is bounded in expectation as*

$$N = \widetilde{O}\left(\max\left\{\kappa d^{1/2}(d \vee \beta) \log^3 \tfrac{\phi}{\varepsilon}, \tfrac{\kappa d^{3/2}\phi}{\varepsilon^2}\right\}\right).$$

*Remark* 4.15. It can be seen that using $d$ i.i.d. samples would require $\widetilde{O}\left(\frac{\kappa d^{3/2}(d \vee \beta)\phi}{\varepsilon^2}\right)$ queries in expectation. This is always worse than the rate above by a factor of at least $d$. Note that Theorem 4.14 also implies a concentration result for the mean, and a similar proof will yield guarantees resembling that of Theorem 3.4, albeit with additional terms in the query complexity.

## 5. Concluding remarks

In summary, we have established in this work that Markov chains can estimate the covariance of a target distribution under the joint assumptions of (PI) and a spectral gap. We demonstrated two applications where this has a substantial benefit in terms of query complexity bounds. In particular, we demonstrated its relevance to an iterative rounding algorithm, which is a cornerstone of the literature for convex body sampling.

The guarantees are conditional on the availability of a useful Poincaré constant and a sampler whose iterated kernel has a spectral gap. These assumptions hold for several log-concave and convex-body settings, but they do not cover targets with severe multimodality or samplers for which no spectral-gap estimate is known. Estimating these constants from data is a separate problem.

We foresee several worthwhile extensions to our efforts. In the proof of Lemma A.8, we only obtained a polynomial failure-probability decay for the empirical tail contribution via Markov's inequality due to dependence of samples, whereas Adamczak et al. (2010) was able to find a subexponential bound using independence. Closing this gap would render our result completely analogous with the prior literature in the i.i.d. setting. An interesting complementary approach may be to establish a Poincaré inequality or other functional inequality directly on the joint distribution of the iterates $\{X_i\}_{i \leq N}$. Other open directions include matching lower bounds for Markov-dependent covariance estimation, discrete-state analogues based on discrete Poincaré inequalities, and extensions to generalized Dirichlet forms with the tail geometry induced by the corresponding carré du champ.

## Impact Statement

This paper presents work whose goal is to advance the field of Machine Learning. There are many potential societal consequences of our work, none of which we feel must be specifically highlighted here.

## Acknowledgements

YK was supported in part by NSF Award CCF-2106444. MSZ was supported in part by an NSERC CGS-D award.

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

# A. Details for the main results

## A.1. Matrix Bernstein inequality under spectral gap of a Markov chain

*Proof of Corollary 3.3.* We deduce this from Theorem 3.2. Consider the matrix

$$F(X_i) := \begin{bmatrix} 0 & X_i^{\mathsf{T}} \\ X_i & 0_{d \times d} \end{bmatrix}.$$

Clearly, $\mathbb{E}[F(X_j)] = 0$. Next, its operator norm is bounded as follows: for $(v, w) \in \mathbb{R} \times \mathbb{R}^d$ with $\|(v, w)\| = 1$,

$$\|F(X_j)\| = 2 \sup_{\|(v,w)\|=1} v \, X_j^{\mathsf{T}} w \le 2 \sup |v| \|w\| \|X_j\| \le \|X_j\| \le \mathcal{M},$$

and the supremum is achieved by $v = 1/\sqrt{2}$ and $w = X_j/(\sqrt{2}\|X_j\|)$. Also, observe that

$$\mathbb{E}[F(X_j)^2] = \begin{bmatrix} \mathbb{E}[\|X_j\|^2] & 0 \\ 0 & \mathbb{E}[X_j X_j^{\mathsf{T}}] \end{bmatrix} \preceq \mathbb{E}[\|X_j\|^2] \, I_{d+1},$$

where the inequality follows from $\mathbb{E}[X_j X_j^{\mathsf{T}}] \preceq \mathbb{E}[\|X_j\|^2] \, I_d$. Therefore, $\|\mathbb{E}[F(X_j)^2]\| \le \mathbb{E}[\|X_j\|^2] \le \mathcal{V}$.

Noting that the operator norm of

$$\sum_{j=1}^{n} F(X_j) = \begin{bmatrix} 0 & \sum X_j^{\mathsf{T}} \\ \sum X_j & 0 \end{bmatrix}$$

is $\|\sum X_j\|$, and using Theorem 3.2,

$$\mathbb{P}\Big(\Big\|\sum_{j=1}^{n} X_j\Big\| \ge t\Big) = \mathbb{P}\Big(\Big\|\sum_{j=1}^{n} F(X_j)\Big\| \ge t\Big) \le (d+1)^{2-\pi/4} \exp\Big(\frac{-t^2/(32/\pi^2)}{\alpha(\lambda)\,\sigma^2 + \beta(\lambda)\,\mathcal{M}t}\Big),$$

which completes the proof. $\qquad\square$

The following lemma allows us to perform most of our calculations at stationarity, paying only a small overhead.

**Lemma A.1** (Change of measure). *Suppose a Markov chain with kernel $P$ starts from $\nu$ instead of the stationary measure $\pi$. Then, the probability of an event has either a multiplicative factor of $\|\frac{\mathrm{d}\nu}{\mathrm{d}\pi}\|_{\infty}$, or, if one is willing to trade-off the failure probability, a Cauchy–Schwarz bound*

$$\nu_{1:n}(A) \le \sqrt{1 + \chi^2(\nu_1 \,\|\, \pi_1)} \sqrt{\pi_{1:n}(A)}.$$

*Proof.* Let $A$ be the bad event we want to bound. Let us denote $\pi_{1:n} := \mathrm{law}(s_1, \ldots, s_n)$ for $s_1 \sim \pi$ and $\nu_{1:n} := \mathrm{law}(s_1, \ldots, s_n)$ for $s_1 \sim \nu$, where $s_{i+1} \sim P(\cdot|s_i)$.

$\chi^2$ **bound:** We have

$$\nu_{1:n}(A) = \int \mathbb{1}_A \, \mathrm{d}\nu_{1:n} = \int \mathbb{1}_A \frac{\mathrm{d}\nu_{1:n}}{\mathrm{d}\pi_{1:n}} \, \mathrm{d}\pi_{1:n} \le \sqrt{\big(1 + \chi^2(\nu_{1:n} \,\|\, \pi_{1:n})\big)} \cdot \sqrt{\pi_{1:n}(A)}.$$

Note that

$$\begin{aligned}
1 + \chi^2(\nu_{1:n} \,\|\, \pi_{1:n}) &= \int \Big(\frac{\nu_{1:n}}{\pi_{1:n}}\Big)^2 \mathrm{d}\pi_{1:n} = \int \int \Big(\frac{\nu_{1:n-1}}{\pi_{1:n-1}}\Big)^2 \Big(\frac{\nu_{n|-n}}{\pi_{n|-n}}\Big)^2 \mathrm{d}\pi_{n|-n} \mathrm{d}\pi_{1:n-1} \\
&\underset{(i)}{=} \int \int \Big(\frac{\nu_{1:n-1}}{\pi_{1:n-1}}\Big)^2 \Big(\frac{\nu_{n|n-1}}{\pi_{n|n-1}}\Big)^2 \mathrm{d}\pi_{n|n-1} \mathrm{d}\pi_{1:n-1} \\
&\underset{(ii)}{=} \int \Big(\frac{\nu_{1:n-1}}{\pi_{1:n-1}}\Big)^2 \mathrm{d}\pi_{1:n-1} \\
&\underset{(iii)}{=} \int \Big(\frac{\nu_1}{\pi_1}\Big)^2 \mathrm{d}\pi_1 = 1 + \chi^2(\nu_1 \,\|\, \pi_1),
\end{aligned}$$

where $(i)$ follows from $P$ being a Markov chain, $(ii)$ follows from $\nu_{n|n-1} = \pi_{n|n-1}$, and $(iii)$ follows from induction.

$\|\cdot\|_\infty$ **bound:** Instead of applying Cauchy–Schwarz, we simply obtain

$$\nu_{1:n}(A) \le \|\frac{\mathrm{d}\nu_{1:n}}{\mathrm{d}\pi_{1:n}}\|_\infty \cdot \pi_{1:n}(A) \,.$$

We can reduce the $\|\cdot\|_\infty$ quantity to one between $\nu_1$ and $\pi_1$ as in the $\chi^2$ case. $\qquad\square$

## A.2. Moments under Poincaré

We can establish concentration of the target measure around its mean.

**Lemma A.2.** *Suppose that $\pi \in \mathcal{P}(\mathbb{R}^d)$ satisfies* (PI). *Then, letting $\mu \in \mathbb{R}^d$ denote the mean of $\pi$, for all $t \ge 1$ we have*

$$\mathbb{P}(\|X - \mu\| \ge t\sqrt{\operatorname{tr}\Sigma}) \le \exp\Big(-(t-1)\,\big(\frac{\operatorname{tr}\Sigma}{C_{\mathsf{PI}}(\pi)}\big)^{1/2}\Big) \,.$$

*Proof.* We note that $f(x) = \|x - \mu\|$ is 1-Lipschitz. Hence, it follows from the Lipschitz concentration property in (2.1) that for all $t \ge 0$,

$$\mathbb{P}(\|X - \mu\| \ge t + \mathbb{E}\|X - \mu\|) \le 3\exp\big(-\frac{t}{\sqrt{C_{\mathsf{PI}}(\pi)}}\big) \,.$$

Since $\mathbb{E}\|X - \mu\| \le (\mathbb{E}[\|X - \mu\|^2])^{1/2} = \sqrt{\operatorname{tr}\Sigma}$, we can deduce that

$$\mathbb{P}(\|X - \mu\| \ge t + \sqrt{\operatorname{tr}\Sigma}) \le 3\exp\big(-\frac{t}{\sqrt{C_{\mathsf{PI}}(\pi)}}\big) \,,$$

which completes the proof. $\qquad\square$

One can also bound the fourth-moment under the Poincaré inequality:

**Lemma A.3.** *Suppose that $\pi \in \mathcal{P}(\mathbb{R}^d)$ satisfies* (PI). *Then,*

$$\mathbb{E}[\|X - \mu\|^4] \le \big(4C_{\mathsf{PI}}(\pi) + \operatorname{tr}\Sigma\big)\operatorname{tr}\Sigma \,.$$

*Proof.* Let us assume that $\mu = 0$ by translation. Taking $f(x) := \|x\|^2$ in (PI), we have from $\nabla f(x) = 2x$ and $\mathbb{E}[\|X\|^2] = \operatorname{tr}\Sigma$ that

$$\mathbb{E}[\|X\|^4] \le 4C_{\mathsf{PI}}(\pi)\,\mathbb{E}[\|X\|^2] + (\mathbb{E}[\|X\|^2])^2 = \operatorname{tr}\Sigma\,\big(4C_{\mathsf{PI}}(\pi) + \operatorname{tr}\Sigma\big) \,,$$

which completes the proof. $\qquad\square$

## A.3. Proof details

### A.3.1. MEAN ESTIMATION

The proof of the mean estimation result closely mirrors that of the succeeding covariance estimation bound.

**Lemma A.4.** *Under Assumption 3.1, if $\lambda \ge 0.99$, then with probability at least $1 - \wp/d$ we have that for any $\varepsilon \in (0, d)$,*

$$\|\overline{X} - \mu\|^2 \le \varepsilon \,,$$

*so long as $N \asymp \frac{\operatorname{tr}\Sigma + C_{\mathsf{PI}}(\pi)}{\varepsilon}\log^2 d\,\log^2\frac{d\max(d,\operatorname{tr}\Sigma)}{\varepsilon}$.*

*Proof.* By translation, we may assume that $\pi$ is centered (i.e., $\mu = \mathbb{E}_\pi X = 0$). Let $B_t := \{x \in \mathbb{R}^d : \|x\| \le t\sqrt{\operatorname{tr}\Sigma}\}$, where $t > 0$ are parameters to be determined, and we drop the subscript $t$ where there is no confusion. Then, we have the following decomposition:

$$\frac{1}{N}\sum X_i = \underbrace{\frac{1}{N}\sum(X_i\mathbb{1}_B - \mathbb{E}[X_i\mathbb{1}_B])}_{=:\mathsf{B}_1} + \underbrace{\frac{1}{N}\sum X_i\mathbb{1}_{B^c}}_{=:\mathsf{B}_2} - \underbrace{\mathbb{E}[X\mathbb{1}_{B^c}]}_{=:\mathsf{B}_3} \,.$$

**Term** $B_1$**:** As for this term, we essentially work with the truncated variables $X_i \mathbb{1}_B$. Let $F_i(X_i) = \frac{1}{N}(X_i \mathbb{1}_B - \mathbb{E}[X_i \mathbb{1}_B])$ for some parameter $\lambda > 0$. Clearly, $\mathbb{E}[F_i(X_i)] = 0$, and

$$\|F_i(X_i)\| \leq \frac{1}{N}\|X_i \mathbb{1}_B\| + \frac{1}{N}\mathbb{E}\|X_i \mathbb{1}_B\| \leq \frac{2}{N} t \sqrt{\operatorname{tr}\Sigma}\,.$$

For the variance, using the identity $(a+b)^2 \leq 2(a^2 + b^2)$ for $a, b \in \mathbb{R}$,

$$\mathbb{E}[\|F_i(X_i)\|^2] \leq \frac{2}{N^2}\mathbb{E}[\|X_i \mathbb{1}_B\|^2] + \frac{2}{N^2}\|\mathbb{E}[X_i \mathbb{1}_B]\|^2$$

$$\leq \frac{4}{N^2}\mathbb{E}[\|X_i \mathbb{1}_B\|^2] \leq \frac{4t^2 \operatorname{tr}\Sigma}{N^2}\,.$$

Hence, we can substitute $\mathcal{M} = \frac{2t\sqrt{\operatorname{tr}\Sigma}}{N}$, $\mathcal{V}_i = \frac{4t^2 \operatorname{tr}\Sigma}{N^2}$, and $\sigma^2 = \frac{4t^2 \operatorname{tr}\Sigma}{N}$ into Corollary 3.3, finding that for $\alpha(\lambda) = \frac{2-\lambda}{\lambda}$ and $\beta(\lambda) = \frac{8/\pi}{\lambda}$,

$$\mathbb{P}\Big(\Big\|\frac{1}{N}\sum_{i=1}^{N} X_i \mathbb{1}_B - \mathbb{E}[X_i \mathbb{1}_B]\Big\| \geq \ell\Big) \leq (d+1)^{2-\pi/4} \exp\Big(-\frac{\pi^2 \ell^2/32}{\alpha(\lambda)\sigma^2 + \beta(\lambda)\mathcal{M}\ell}\Big)\,.$$

Since $\alpha(\lambda)$ and $\beta(\lambda)$ are absolute constants, we have that with probability $1 - \mathbb{P}/d$,

$$\Big\|\frac{1}{N}\sum_{i=1}^{N} X_i \mathbb{1}_B - \mathbb{E}[X_i \mathbb{1}_B]\Big\| \lesssim (\mathcal{M}+\sigma)\log d \lesssim \frac{t\sqrt{\operatorname{tr}\Sigma}\log d}{\sqrt{N}}\,.$$

**Terms** $B_2$ **and** $B_3$**:** As for $B_2$, we use Markov's inequality,

$$\mathbb{P}\Big(\Big\|\frac{1}{N}\sum X_i \mathbb{1}_{B^c}\Big\| \geq s\,\mathbb{E}\Big[\Big\|\frac{1}{N}\sum X_i \mathbb{1}_{B^c}\Big\|\Big]\Big) \leq \frac{1}{s} \qquad \text{for any } s > 0\,.$$

Using Lemma A.2 for $t \geq 2$ in the second last line below,

$$\mathbb{E}\Big[\Big\|\frac{1}{N}\sum X_i \mathbb{1}_{B^c}\Big\|\Big] \leq \mathbb{E}\Big[\frac{1}{N}\sum_{i=1}^{N}\|X_i\|\mathbb{1}_{B^c}\Big] = \mathbb{E}_\pi[\|X\|\mathbb{1}_{B^c}] \leq \sqrt{\mathbb{E}[\|X\|^2]}\sqrt{\mathbb{P}(\|X\| \geq t\sqrt{\operatorname{tr}\Sigma})}$$

$$\leq 3\sqrt{\operatorname{tr}\Sigma} \cdot \exp\Big(-\frac{(t-1)\sqrt{\operatorname{tr}\Sigma}}{\sqrt{C_{\mathsf{PI}}(\pi)}}\Big) \leq 3\sqrt{\operatorname{tr}\Sigma}\exp\Big(-\frac{t}{2}\Big(\frac{\operatorname{tr}\Sigma}{C_{\mathsf{PI}}(\pi)}\Big)^{1/2}\Big)\,.$$

This bounds the norm of Term $B_3$ as well by the same argument, since

$$\|\mathbb{E}[X\mathbb{1}_{B^c}]\| \leq \mathbb{E}[\|X\|\mathbb{1}_{B^c}] \leq 3\sqrt{\operatorname{tr}\Sigma}\exp\Big(-\frac{t}{2}\Big(\frac{\operatorname{tr}\Sigma}{C_{\mathsf{PI}}(\pi)}\Big)^{1/2}\Big)\,.$$

Combining all bounds on the three terms, it follows that with probability $1 - \mathbb{P}/s - \mathbb{P}/d$,

$$\Big\|\frac{1}{N}\sum_{i=1}^{N} X_i\Big\| \lesssim \frac{t\sqrt{\operatorname{tr}\Sigma}\log d}{\sqrt{N}} + (s+1)\sqrt{\operatorname{tr}\Sigma}\exp\Big(-\frac{t}{2}\Big(\frac{\operatorname{tr}\Sigma}{C_{\mathsf{PI}}(\pi)}\Big)^{1/2}\Big)\,.$$

Taking $s = 8d$ and $t \asymp \Big(\frac{\operatorname{tr}\Sigma + C_{\mathsf{PI}}(\pi)}{\operatorname{tr}\Sigma}\Big)^{1/2}\log\frac{d\max(d,\operatorname{tr}\Sigma)}{\varepsilon}$, we have

$$\Big\|\frac{1}{N}\sum_{i=1}^{N} X_i\Big\| \lesssim \varepsilon^{1/2} + \Big(\frac{\operatorname{tr}\Sigma + C_{\mathsf{PI}}(\pi)}{N}\Big)^{1/2}\log d \log\frac{d^2 \operatorname{tr}\Sigma}{\varepsilon}\,.$$

Therefore, it suffices to take $N \asymp \frac{\operatorname{tr}\Sigma + C_{\mathsf{PI}}(\pi)}{\varepsilon}\log^2 d \log^2\frac{d\max(d,\operatorname{tr}\Sigma)}{\varepsilon}$ and then revert the translation for centering. $\qquad\square$

For the multiplicative version, we can just work with the relative error $\varepsilon \|\Sigma\|$ in place of $\varepsilon$.

**Corollary A.5.** *Under Assumption 3.1, if $\lambda \geq 0.99$, then with probability at least $1 - \mathbb{P}/d$ we have that for any $\varepsilon \in (0, d)$,*

$$\|\overline{X} - \mu\|^2 \leq \varepsilon \|\Sigma\|\,,$$

*so long as $N \asymp \frac{d}{\varepsilon}\Big(1 \vee \frac{C_{\mathsf{PI}}(\pi)}{\operatorname{tr}\Sigma}\log^2\frac{d}{\varepsilon}\Big)\log^2 d$.*

A.3.2. Covariance estimation

**Additive version.** We extend Jia et al. (2021, Lemma A.2), proven for independent samples distributed according to any log-concave distribution. The following lemma contains the bound for the covariance part of our problem.

**Lemma A.6.** *Under Assumption 3.1, if $\lambda \geq 0.99$, then with probability at least $1 - \textsc{p}/d$ we have*

$$\left| \frac{1}{N} \sum (X_i - \mu)^{\otimes 2} - \Sigma \right| \preceq \varepsilon \Sigma + \delta I$$

*so long as $N \asymp \frac{\operatorname{tr} \Sigma + C_{\mathsf{PI}}(\pi)}{\varepsilon \delta} \log^2 \frac{d \max(d, C_{\mathsf{PI}}(\pi) + \operatorname{tr} \Sigma)}{\delta} \log^2 d.$*

In the analysis, we can make $\pi$ centered (i.e., $\mu = 0$) by translation and work with a truncated region $B_{\rho,t} := \{x \in \mathbb{R}^d : \|x\|_{(\rho\Sigma+I)^{-1}} \leq t\sqrt{\operatorname{tr} \Sigma}\}$ with $t$ and $\rho$ to be determined; we again suppress the subscripts. Recall from §3 that we have the following decomposition:

$$\frac{1}{N} \sum X_i^{\otimes 2} - \Sigma = \underbrace{\frac{1}{N} \sum (X_i^{\otimes 2} \mathbb{1}_B - \mathbb{E}[X_i^{\otimes 2} \mathbb{1}_B])}_{=:A_1} + \underbrace{\frac{1}{N} \sum X_i^{\otimes 2} \mathbb{1}_{B^c}}_{=:A_2} - \underbrace{\mathbb{E}[X^{\otimes 2} \mathbb{1}_{B^c}]}_{=:A_3}.$$

We now bound each term separately.

**Lemma A.7** (Term $A_1$). *With probability at least $1 - \textsc{p}/d$, for $t \geq 2$,*

$$\left| \frac{1}{N} \sum (X_i^{\otimes 2} \mathbb{1}_B - \mathbb{E}[X_i^{\otimes 2} \mathbb{1}_B]) \right| \preceq \varepsilon \Sigma + O\left( \frac{t^2 \operatorname{tr} \Sigma \log^2 d}{\varepsilon N} \right) I_d.$$

*Proof.* We use Theorem 3.2, setting $F_i(X_i) = \frac{1}{N}(\rho\Sigma+I)^{-1/2}(X_i^{\otimes 2}\mathbb{1}_B - \mathbb{E}[X_i^{\otimes 2}\mathbb{1}_B])(\rho\Sigma+I)^{-1/2}$ for $i \in [N]$. Denoting $X_i^{\otimes 2}/(\rho\Sigma+I) := (\rho\Sigma+I)^{-1/2}X_i^{\otimes 2}(\rho\Sigma+I)^{-1/2}$, we have

$$
\begin{aligned}
\|F_i(X_i)\| &\leq \frac{\mathbb{1}_B}{N} \left\| \frac{X_i^{\otimes 2}}{\rho\Sigma+I} \right\| + \frac{1}{N}\mathbb{E}\left[ \left\| \frac{X_i^{\otimes 2}}{\rho\Sigma+I} \right\| \mathbb{1}_B \right] \\
&= \frac{1}{N}\|X_i\|_{(\rho\Sigma+I)^{-1}}^2 \mathbb{1}_B + \frac{1}{N}\mathbb{E}[\|X_i\|_{(\rho\Sigma+I)^{-1}}^2 \mathbb{1}_B] \\
&\leq \frac{2}{N}t^2 \operatorname{tr} \Sigma,
\end{aligned}
$$

where the last line follows from the definition of $B$. Hence, we set $\mathcal{M} = 2t^2 \operatorname{tr} \Sigma/N$ in the theorem. As for the variance,

$$
\begin{aligned}
\mathbb{E}[F_i(X_i)^2] &\underset{(i)}{\preceq} \frac{2}{N^2}\mathbb{E}\left[ \left( \frac{X_i^{\otimes 2}}{\rho\Sigma+I} \right)^2 \mathbb{1}_B \right] + \frac{2}{N^2}\left( \mathbb{E}\left[ \frac{X_i^{\otimes 2}}{\rho\Sigma+I} \mathbb{1}_B \right] \right)^2 \\
&\underset{(ii)}{\preceq} \frac{2}{N^2}\mathbb{E}\left[ \|X_i\|_{(\rho\Sigma+I)^{-1}}^2 \frac{X_i^{\otimes 2}}{\rho\Sigma+I} \mathbb{1}_B \right] + \frac{2}{N^2}\left\| \mathbb{E}\left[ \frac{X_i^{\otimes 2}}{\rho\Sigma+I} \mathbb{1}_B \right] \right\| \mathbb{E}\left[ \frac{X_i^{\otimes 2}}{\rho\Sigma+I} \mathbb{1}_B \right] \\
&\underset{(iii)}{\preceq} \frac{2t^2 \operatorname{tr} \Sigma}{N^2} \frac{\Sigma}{\rho\Sigma+I} + \frac{2t^2 \operatorname{tr} \Sigma}{N^2} \frac{\Sigma}{\rho\Sigma+I} = \frac{2\mathcal{M}}{N} \frac{\Sigma}{\rho\Sigma+I},
\end{aligned}
$$

where in $(i)$ we used $(A - B)^2 \preceq 2(A^2 + B^2)$ for matrices $A$ and $B$, in $(ii)$ used $A^2 \preceq \|A\|A$ for a positive semidefinite matrix $A$, and $(iii)$ follows from $\mathbb{E}[X_i^{\otimes 2}] = \Sigma$ and the definition of $B$. Hence,

$$\|\mathbb{E}[F_i(X_i)^2]\| \leq \frac{2\mathcal{M}}{N} \left\| \frac{\Sigma}{\rho\Sigma+I} \right\|,$$

so we can set $\mathcal{V}_i = 2\mathcal{M}\|\Sigma/(\rho\Sigma+I)\|/N$ and $\sigma^2 = 2\mathcal{M}\|\Sigma/(\rho\Sigma+I)\|$.

Using Theorem 3.2 with $\alpha(\lambda), \beta(\lambda) = O(1)$, we have

$$\mathbb{P}\left( \left\| \frac{1}{N}(\rho\Sigma+I)^{-1/2} \sum (X_i^{\otimes 2}\mathbb{1}_B - \mathbb{E}[X_i^{\otimes 2}\mathbb{1}_B])(\rho\Sigma+I)^{-1/2} \right\| \geq \ell \right)$$

$$\le d2^{-\pi/4}\exp\Big(-\frac{\ell^2/(32/\pi^2)}{\alpha(\lambda)\,\sigma^2+\beta(\lambda)\,\mathcal{M}\ell}\Big)\,.$$

Hence, with probability at least $1-\mathbb{P}/d$, using Young's inequality with any $c>0$,

$$\|(\rho\Sigma+I)^{-1/2}\mathsf{A}_1(\rho\Sigma+I)^{-1/2}\|\lesssim(\mathcal{M}+\sigma)\log d\le\Big(\mathcal{M}+\sqrt{2\mathcal{M}\Big\|\frac{\Sigma}{\rho\Sigma+I}\Big\|}\Big)\log d$$

$$\lesssim\Big((1+c)\mathcal{M}+\frac{1}{c}\Big\|\frac{\Sigma}{\rho\Sigma+I}\Big\|\Big)\log d$$

$$\le\Big((1+c)\frac{t^2\operatorname{tr}\Sigma}{N}+\frac{1}{c\rho}\Big)\log d\,.$$

Therefore, with probability at least $1-\mathbb{P}/d$,

$$\mathsf{A}_1\preceq\Big((1+c)\frac{t^2\operatorname{tr}\Sigma}{N}+\frac{1}{c\rho}\Big)\log d\cdot(\rho\Sigma+I_d)\,.$$

By solving $\frac{\log d}{c}=\varepsilon$ and $\frac{ct^2\operatorname{tr}\Sigma}{N}=\frac{1}{c\rho}$ for $c$ and $\rho$, there exist multiples of $c$ and $\rho$ such that

$$\mathsf{A}_1\preceq\varepsilon\Sigma+O\Big(\frac{t^2\operatorname{tr}\Sigma\log^2 d}{\varepsilon N}\Big)I_d\,.$$

Under the same event, we also have $-\mathsf{A}_1\preceq\varepsilon\Sigma+O(\frac{t^2\operatorname{tr}\Sigma\log^2 d}{\varepsilon N})I_d$ in a similar manner. $\qquad\square$

**Lemma A.8** (Term $\mathsf{A}_2$ and $\mathsf{A}_3$). *For $s>0$ and $t\ge 2$, with probability at least $1-\mathbb{P}/s$,*

$$\Big\|\frac{1}{N}\sum X_i^{\otimes 2}\mathbb{1}_{B^c}\Big\|\le s\,\mathbb{E}[X^{\otimes 2}\mathbb{1}_{B^c}]\le 3s\sqrt{(4C_{\mathsf{PI}}(\pi)+\operatorname{tr}\Sigma)\operatorname{tr}\Sigma}\exp\Big(-\frac{t}{2}\Big(\frac{\operatorname{tr}\Sigma}{C_{\mathsf{PI}}(\pi)}\Big)^{1/2}\Big)\,.$$

*Proof.* Using Markov's inequality, we have

$$\mathbb{P}\Big(\Big\|\frac{1}{N}\sum X_i^{\otimes 2}\mathbb{1}_{B^c}\Big\|\ge s\,\mathbb{E}\Big[\Big\|\frac{1}{N}\sum X_i^{\otimes 2}\mathbb{1}_{B^c}\Big\|\Big]\Big)\le\frac{1}{s}\qquad\text{for any }s>0\,.$$

The expectation can be bounded by

$$\mathbb{E}\Big[\Big\|\frac{1}{N}\sum X_i^{\otimes 2}\mathbb{1}_{B^c}\Big\|\Big]\le\mathbb{E}\Big[\frac{1}{N}\sum_i\|X_i\|^2\mathbb{1}_{B^c}\Big]=\mathbb{E}[\|X\|^2\mathbb{1}_{B^c}]$$

$$\le\sqrt{\mathbb{E}[\|X\|^4]}\sqrt{\mathbb{P}\big(\|X\|\ge t\sqrt{\operatorname{tr}\Sigma}\big)}$$

$$\le 3\sqrt{(4C_{\mathsf{PI}}(\pi)+\operatorname{tr}\Sigma)\operatorname{tr}\Sigma}\exp\Big(-\frac{t}{2}\Big(\frac{\operatorname{tr}\Sigma}{C_{\mathsf{PI}}(\pi)}\Big)^{1/2}\Big)\qquad\text{for any }t\ge 2\,,$$

where the bounds on the first and the second term follow from Lemma A.3 and Lemma A.2 respectively. Since $\|\mathsf{A}_3\|=\|\mathbb{E}[X^{\otimes 2}\mathbb{1}_{B^c}]\|\le\mathbb{E}[\|X\|^2\mathbb{1}_{B^c}]$, it is also bounded by the last bound above. $\qquad\square$

*Proof of Lemma A.6.* Putting these bounds together with $s=d$, with probability at least $1-\mathbb{P}/d$,

$$\Big|\frac{1}{N}\sum X_i^{\otimes 2}-\Sigma\Big|\preceq\varepsilon\Sigma+O\Big(\frac{t^2\operatorname{tr}\Sigma\log^2 d}{\varepsilon N}+\underbrace{d\,(C_{\mathsf{PI}}(\pi)+\operatorname{tr}\Sigma)\exp\Big(-\frac{t}{2}\Big(\frac{\operatorname{tr}\Sigma}{C_{\mathsf{PI}}(\pi)}\Big)^{1/2}\Big)}_{=:(\#)}\Big)I_d\,.$$

By setting $t\asymp 2(\frac{\operatorname{tr}\Sigma+C_{\mathsf{PI}}(\pi)}{\operatorname{tr}\Sigma})^{1/2}\log\frac{d\max(d,C_{\mathsf{PI}}(\pi)+\operatorname{tr}\Sigma)}{\delta}$, we can make $(\#)\lesssim\delta$. The claim then follows by taking

$$N\asymp\frac{\operatorname{tr}\Sigma+C_{\mathsf{PI}}(\pi)}{\varepsilon\delta}\log^2 d\log^2\frac{d\max(d,C_{\mathsf{PI}}(\pi)+\operatorname{tr}\Sigma)}{\delta}\,,$$

which completes the proof. $\qquad\square$

*Proof of Theorem 3.4.* Consider $\Sigma' = \frac{1}{N}\sum_i (X_i - \mu)^{\otimes 2} = \frac{1}{N}\sum_i X_i^{\otimes 2} - \overline{X}\mu^\mathsf{T} - \mu\overline{X}^\mathsf{T} + \mu\mu^\mathsf{T}$, where $\mu$ is the true mean $\mathbb{E}_\pi X$. Now, we have

$$\overline{\Sigma} - \Sigma' = \frac{1}{N}\sum_{i=1}^N X_i^{\otimes 2} - \overline{X}^{\otimes 2} - \Big(\frac{1}{N}\sum_{i=1}^N X_i^{\otimes 2} - \overline{X}\mu^\mathsf{T} - \mu\overline{X}^\mathsf{T} + \mu\mu^\mathsf{T}\Big) = -(\overline{X} - \mu)^{\otimes 2}.$$

For the true covariance $\Sigma = \mathbb{E}[X^{\otimes 2}] - \mu^{\otimes 2}$, observe that

$$\overline{\Sigma} - \Sigma = \overline{\Sigma} - \Sigma' + \Sigma' - \Sigma = \underbrace{\frac{1}{N}\sum_{i=1}^N\big((X_i - \mu)^{\otimes 2} - \Sigma\big)}_{=:\mathsf{A}} - \underbrace{(\overline{X} - \mu)^{\otimes 2}}_{=:\mathsf{B}}.$$

It suffices therefore to separately control the error for the centred covariance estimator in A (Lemma A.6), and the mean estimation error in B (Lemma A.4).

The theorem then follows immediately upon combining Lemma A.4 and A.6. $\qquad\square$

**Multiplicative version.** The proof of this version is similar in overall with the previous one, proceeding with a truncated region $B_t := \{x \in \mathbb{R}^d : \|x\| \le t\sqrt{\operatorname{tr}\Sigma}\}$ instead of $B_{\rho,t}$.

**Lemma A.9.** *Under Assumption 3.1, if $\lambda \ge 0.99$, then with probability at least $1 - \mathbb{P}/d$ we have*

$$\Big\|\frac{1}{N}\sum X_i^{\otimes 2} - \Sigma\Big\| \le \varepsilon\|\Sigma\|$$

*so long as $N \asymp \frac{d}{\varepsilon^2}\frac{\operatorname{tr}\Sigma + C_{\mathsf{PI}}(\pi)}{\operatorname{tr}\Sigma}\log^2\frac{d\,(C_{\mathsf{PI}}(\pi)+\operatorname{tr}\Sigma)}{\varepsilon\|\Sigma\|}\log^2 d.$*

*Proof.* Following the proof of Lemma A.7 with $\rho = 0$, we have that with probability at least $1 - \mathbb{P}/d$, using Young's inequality with $c = \frac{\log d}{\varepsilon}$,

$$\|\mathsf{A}_1\| \lesssim \Big((1+c)\frac{t^2\operatorname{tr}\Sigma}{N} + \frac{1}{c}\|\Sigma\|\Big)\log d \le \varepsilon\|\Sigma\| + \frac{t^2\operatorname{tr}\Sigma\log^2 d}{\varepsilon N}.$$

Combining this with the bounds for terms $\mathsf{A}_2$ and $\mathsf{A}_3$ in Lemma A.8, it follows that

$$\Big\|\frac{1}{N}\sum X_i^{\otimes 2} - \Sigma\Big\| \lesssim \varepsilon\|\Sigma\| + \frac{t^2\operatorname{tr}\Sigma\log^2 d}{\varepsilon N} + d\big(C_{\mathsf{PI}}(\pi) + \operatorname{tr}\Sigma\big)\exp\Big(-\frac{t}{2}\Big(\frac{\operatorname{tr}\Sigma}{C_{\mathsf{PI}}(\pi)}\Big)^{1/2}\Big).$$

We can bound the third term by $\varepsilon\|\Sigma\|$ by setting $t \ge 2 \vee \big(\frac{C_{\mathsf{PI}}(\pi)}{\operatorname{tr}\Sigma}\big)^{1/2}\log\frac{d(C_{\mathsf{PI}}(\pi)+\operatorname{tr}\Sigma)}{\varepsilon\|\Sigma\|}$. Under this choice $t$, the second term can be bounded by $\varepsilon\|\Sigma\|$ if we take

$$N \asymp \frac{t^2 d\log^2 d}{\varepsilon^2} \asymp \frac{d}{\varepsilon^2}\Big(1 \vee \frac{C_{\mathsf{PI}}(\pi)}{\operatorname{tr}\Sigma}\log^2\frac{d\big(C_{\mathsf{PI}}(\pi) + \operatorname{tr}\Sigma\big)}{\varepsilon\|\Sigma\|}\Big)\log^2 d,$$

which completes the proof. $\qquad\square$

*Proof of Theorem 3.5.* As in the proof of Theorem 3.4, we just combine the two bounds (i.e., errors for the mean and covariance) from Corollary A.5 and Lemma A.9. $\qquad\square$

*Proof of Corollary 3.6.* We just apply Theorem 3.5 after transforming the whole system by $x \mapsto \Sigma^{-1/2}x$. Then, the covariance becomes $\Sigma = I$, and we obtain that for $N \asymp \frac{d + C_{\mathsf{PI}}(\nu)}{\varepsilon^2}\log^2 d\,\log^2\frac{d\,(d + C_{\mathsf{PI}}(\nu))}{\varepsilon}$,

$$|\Sigma^{-1/2}\overline{\Sigma}\Sigma^{-1/2} - I| \preceq \varepsilon I_d,$$

from which the claim follows by conjugating both sides by $\Sigma^{1/2}$. $\qquad\square$

# B. Details for the applications

## B.1. Detailed algorithm

In the algorithm, let $\pi_i$ denote $\pi_{\mathcal{K}_i}$, and $\Sigma_i$ be its covariance.

---
**Algorithm 3** Isotropize
---
**Input:** convex body $\mathcal{K} \subset \mathbb{R}^d$ and $T_1 \in \mathbb{R}^{d \times d}$ such that $\mathcal{K}_1 := T_1 \mathcal{K}$ satisfies $1/4 \leq \mathsf{inrad}(\mathcal{K}_1)$ and $\mathbb{E}_{\pi_{\mathcal{K}_1}}[\|\cdot\|^2] \leq C^2 d$ for constant $C > 0$.
**Output:** $(\widehat{\mu}, \widehat{\Sigma}^{-1/2})$.

1: Run Gaussian cooling to obtain $Z_1 \in \mathcal{K}_1$ with $\mathcal{R}_\infty(\mathrm{law}(Z_1) \,\|\, \pi_{\mathcal{K}_1}) \leq \log 2$.
2: Let $r_1 = 1/4$, $\pi_1 := \pi_{\mathcal{K}_1}$, and $i = 1$.
3: **while** $r_i^2 \leq \frac{d}{2^{10} \log^4 d}$ **do**
4:   Set $k_i = 10 c r_i^2 C^2 \log^6(C^2 d)$ for some universal constant $c$.
5:   Draw $\{X_{i,j}\}_{j \in [k_i]} \leftarrow \mathsf{In\text{-}and\text{-}Out}_{N_i}\left(\pi_i, \delta_{Z_i}, \frac{r_i^2}{2^{10} d^2 \log(Ccr_i d)}\right)$ with $N_i = \frac{2^{10} C^2 d^3 \log(Ccd)}{r_i^2}$.
6:   Compute $\widehat{\mu}_i = \frac{1}{k_i} \sum_{j=1}^{k_i} X_{i,j}$ and $\widehat{\Sigma}_i = \frac{1}{k_i} \sum_{j=1}^{k_i} (X_{i,j} - \widehat{\mu}_i)^{\otimes 2}$.
7:   Compute $M_i = I_d + P_i$, where $P_i$ is the orthogonal projection to the subspace spanned by eigenvectors of $\widehat{\Sigma}_i$ with eigenvalue at most $d$.
8:   Set $T_{i+1} = M_i T_i$, $\mathcal{K}_{i+1} = M_i \mathcal{K}_i$, $Z_{i+1} = M_i Z_i$, $r_{i+1} = 2r_i(1 - 1/\log d)$, and $i \leftarrow i + 1$.
9: **end while**
10: Draw $c' d \log^6 d$ outputs from $\mathsf{In\text{-}and\text{-}Out}_N\left(\pi_i, \delta_{Z_i}, \frac{1}{2^{20} d \log(Ccr_i d)}\right)$ with $N = 2^{20} C^2 d^2 \log^5(Ccd)$ (for some universal constant $c'$), and use them to compute the mean $\widehat{\mu}$ and covariance $\widehat{\Sigma}$.
---

## B.2. Isotropic rounding

We show that $\mathsf{In\text{-}and\text{-}Out}_{N_i}(\pi_i, \cdot, h_i)$ has a spectral gap at least $0.99$. The following estimates are used inductively. At the beginning of loop $i$, assume the trace/operator-norm bounds and the inner-radius certificate have been propagated from the previous loops. Lemma 4.2 gives the required spectral gap, Lemma 4.3 gives the covariance event in the current loop, and Lemmas 4.4–4.5 propagate the invariants to the next loop.

*Proof of Lemma 4.2.* Within each while-loop, In-and-Out iterates $k_i N_i = \widetilde{O}(cC^4 d^3)$ times, and the number of while-loops is at most $2 \log d$, as we will show later in Lemma 4.4. Therefore, the total number of iterations of In-and-Out throughout Algorithm 1 is $T := \widetilde{O}(cC^4 d^3)$. With the total failure probability of In-and-Out (throughout the entire algorithm) set to $\eta = 1/d$, Kook et al. (2024, Theorem 27) requires the variance $h$ of In-and-Out to be smaller than $r_i^2 (2d^2 \log \frac{18T}{\eta})^{-1} = r_i^2 (2d^2 \log 18dT)^{-1}$, justifying the choice of $h_i$ in Line 4.

By (4.1) (or Kook et al. (2024, Theorem 23)), it suffices to take $N_i \geq C_0 h_i^{-1} C_{\mathsf{PI}}(\pi_i)$ for a universal constant $C_0$ to ensure the spectral gap is at least $0.99$. Since $C_{\mathsf{PI}}(\pi_i) \lesssim \|\Sigma_i\| \log d$ (Remark 3.7) and $\|\Sigma_i\| \lesssim C^2 d \log d$ (Lemma 4.4), $\mathsf{In\text{-}and\text{-}Out}_{N_i}(\pi_i, \cdot, h_i)$ with $N_i = 2^{10} C^2 d^3 r_i^{-2} \log(Ccd) \log^2 d$ has the desired spectral gap. The choices of $h$ and $N$ in Line 8 can be justified in a similar way. $\qquad\square$

**Covariance estimation.** Recall that Theorem 3.4 assumes that the initial distribution of a Markov chain is already stationary. In the $i$-th loop of Algorithm 1, however, the chain targets $\pi_i$ but starts from the current warm-start law $\nu_i := \mathrm{law}(Z_i)$. We address this in Lemma 4.3; the conclusion is that we only pay a small additional factor in order to handle this initial bias, which can be absorbed into the other terms.

*Proof of Lemma 4.3.* Let $\nu_1 = \mathrm{law}(Z_1)$ be the law of the warm point generated by Gaussian cooling in Line 2. At the beginning of the $i$-th loop, $Z_i = (M_{i-1} \cdots M_1) Z_1$ and $\pi_i = (M_{i-1} \cdots M_1)_\# \pi_{\mathcal{K}_1}$. Hence the $R_\infty$ warmness is invariant under the common affine pushforward:

$$\mathcal{R}_\infty(\mathrm{law}(Z_i) \,\|\, \pi_i) = \mathcal{R}_\infty(\mathrm{law}(Z_1) \,\|\, \pi_{\mathcal{K}_1}) \leq \log 2 \,.$$

Following a similar argument as in Lemma A.1, we note that all the required bounds on bad events in the proof of Theorem 3.4 are multiplied by at most 2, and that $\mathbb{E}_{\nu_i}[\|X\|^q] \leq 3\mathbb{E}_{\pi_i}[\|X\|^q]$ for $q = 2, 4$. Therefore, in the $i$-th loop, we still have that $|\widehat{\Sigma}_i - \Sigma_i| \preceq \varepsilon\Sigma_i + \delta I_d$ for sufficiently many samples $N$.

As shown later in Lemma 4.4, $\operatorname{tr}\Sigma_i \lesssim C^2 d^2$ throughout the algorithm, and it is well-known that $C_{\mathsf{PI}}(\pi_i) \lesssim \operatorname{tr}\Sigma_i$ (due to the log-concavity of $\pi_i$). Hence, with $\varepsilon = 0.1$ and $\delta = d/100$, there exists a universal constant $c$ such that so long as $k_i \geq cd^{-1}\operatorname{tr}\Sigma_i\log^6(C^2d)$,

$$|\widehat{\Sigma}_i - \Sigma_i| \preceq \frac{1}{10}\Sigma_i + \frac{d}{100}I_d\,,$$

which completes the proof. $\qquad\square$

We provide quantitative control of subroutines executed within each while-loop.

*Proof of Lemma 4.4.* The inner radius $r_i$ increases by a factor of at least $3/2$ each iteration. Since the algorithm starts with $r_1 = 1/4$ and ends before $r_j \leq \sqrt{d}$, it takes fewer than $2\log d$ iterations.

Now we analyze how the trace and operator norm of the covariance changes each iteration.

**(1)** As per the algorithm, we have

$$\Sigma_{i+1} = M_i\Sigma_iM_i \tag{B.1}$$

Since $P_i$ is an orthogonal projection matrix,

$$\operatorname{tr}\Sigma_{i+1} = \operatorname{tr}(\Sigma_i^{1/2}M_i^2\Sigma_i^{1/2}) = \operatorname{tr}\bigl(\Sigma_i^{1/2}(I + 3P_i)\Sigma_i^{1/2}\bigr) \leq 4\operatorname{tr}\Sigma_i\,.$$

As $r_{i+1} = 2(1 - \log^{-1}d)\,r_i$, we have that $\operatorname{tr}\Sigma_i/r_i^2$ increases by a factor of $(1 - 1/\log d)^{-2}$ per iteration. Thus, over at most $2\log d$ iterations, this ratio increases by at most a universal constant; absorbing also the factor $r_1^{-2} = 16$, we obtain $\operatorname{tr}\Sigma_i \leq c_{\operatorname{tr}}r_i^2C^2d$ for all $i$.

**(2)** Note that $\|\Sigma_1\| \leq \operatorname{tr}\Sigma_1 \leq C^2d$. Recall that $A^\mathsf{T}A$ and $AA^\mathsf{T}$ share the same spectrum (i.e., the same set of non-zero eigenvalues). Hence, it follows from (B.1) that

$$\|\Sigma_{i+1}\| = \|M_i\Sigma_iM_i\| = \|\Sigma_i^{1/2}M_i^2\Sigma_i^{1/2}\| = \|\Sigma_i^{1/2}(I + 3P_i)\Sigma_i^{1/2}\|$$
$$\leq \|\Sigma_i\| + 3\,\|\Sigma_i^{1/2}P_i\Sigma_i^{1/2}\| = \|\Sigma_i\| + 3\,\|P_i\Sigma_iP_i\|\,. \tag{B.2}$$

Using $k_i = 10cr_i^2C^2\log^6(C^2d)$ many samples, with the universal constant $c$ chosen large enough so that $k_i \geq cd^{-1}\operatorname{tr}\Sigma_i\log^6(C^2d)$ by the trace bound above, Lemma 4.3 ensures that $0.9\Sigma_i - \frac{d}{100}I_d \preceq \widehat{\Sigma}_i \preceq 1.1\Sigma_i + \frac{d}{100}I_d$ with probability at least $1 - \mathbb{P}/d$. Conjugating by the projection matrix $P_i$, we have using a somewhat lazy bound that

$$P_i\Sigma_iP_i \preceq \frac{3}{2}\,P_i\widehat{\Sigma}_iP_i + \frac{d}{50}\,P_i\,.$$

Let $\widehat{\Sigma}_i = U^\mathsf{T}DU$ be its spectral decomposition with orthogonal matrix $U$. Then, under this decomposition, both $UP_iU^\mathsf{T}$ and $D = U\widehat{\Sigma}_iU^\mathsf{T}$ are diagonal, where the entry in $UP_iU^\mathsf{T}$'s diagonal is set to 0 if the corresponding diagonal entry of $U\widehat{\Sigma}_iU^\mathsf{T}$ is larger than $d$, and 1 otherwise. Letting $\min(dI_d, D)$ be the entry-wise minimum of $dI_d$ and $D$, we have that

$$UP_i\Sigma_iP_iU^\mathsf{T} \preceq \frac{3}{2}\,\min(dI_d, D) + \frac{d}{50}I_d \preceq 2dI_d\,,$$

and $\|P_i\Sigma_iP_i\| \leq 2d$. Substituting this back to (B.2), we obtain $\|\Sigma_{i+1}\| \leq \|\Sigma_i\| + 6d$. Therefore, the claim follows from recursing $i$ times. $\qquad\square$

**Doubling of the inner radius.**  We recall an additional technical lemma.

**Lemma B.1** ((Lovász & Vempala, 2006), Lemma 3.4). *Let $\mathcal{K}$ be convex in $\mathbb{R}^d$ with centroid $\mu$ and covariance $\Sigma$ satisfying $r^2I_d \preceq \Sigma \preceq R^2I_d$. Then,*

$$B_r(\mu) \subset \mathcal{K} \subset B_{Rd}(\mu)\,.$$

Lastly, we show that under the update rule of $r_{i+1} \leftarrow 2(1 - 1/\log d)r_i$, the inner radius actually almost doubles if all the covariance estimations thus far have been accurate.

*Proof of Lemma 4.5.* We show that when $B_{r_i}(c_i) \subseteq \mathcal{K}_i$ for some $c_i \in \mathcal{K}_i$, there exists some center $c_{i+1} \in \mathcal{K}_{i+1}$ such that $\mathcal{K}_{i+1}$ contains $B_{r_{i+1}}(c_{i+1})$ with $r_{i+1} = 2(1 - 1/\log d)\, r_i$. By Lemma B.1, $\mathcal{K}_i$ also contains the ellipsoid $\{x : \|x - \mu_i\|^2_{\Sigma_i^{-1}} \leq 1\}$ for the mean $\mu_i$ and covariance $\Sigma_i$ of $\mathcal{K}_i$ with respect to $\pi_i$. Recall that

$$M_i = I + P_i , \quad T_{i+1} = M_i T_i , \quad \Sigma_{i+1} = M_i \Sigma_i M_i .$$

We first focus on the case where the two centers $c_i$ and $\mu_i$ are different.

Let $\widehat{\Sigma}_i = U^\mathsf{T} D_i U$ be the spectral decomposition of the estimated covariance $\widehat{\Sigma}_i$, where $U \in \mathbb{R}^{d \times d}$ is an orthogonal matrix, and $D_i \in \mathbb{R}^{d \times d}$ is a diagonal matrix with eigenvalues on the diagonal in decreasing order. Under the transformation $x \mapsto y := M_i x$ (i.e., $\mathcal{K}_i \to \mathcal{K}_{i+1}$), the new convex body $\mathcal{K}_{i+1}$ contains two ellipsoids: defining $c_i' := M_i c_i$ and $\mu_i' := M_i \mu_i (= \mu_{i+1})$,

$$\mathcal{A} : \{y \in \mathbb{R}^d : (y - c_i')^\mathsf{T}(I + P_i)^{-2}(y - c_i') \leq r_i^2\}, \quad \text{and} \quad \mathcal{B} : \{y \in \mathbb{R}^d : (y - \mu_i')^\mathsf{T}\Sigma_{i+1}^{-1}(y - \mu_i') \leq 1\}.$$

We now work with a new coordinate system in $z := Uy$ for the ease of analysis. Under this new system, there exists $0 \leq m \leq d$ such that the $m$ largest eigenvalues of $U\widehat{\Sigma}_i U^\mathsf{T} = D_i$ (corresponding to bases $\{e_1, \ldots, e_m\}$) are larger than the threshold $d$ and the remaining $d - m$ eigenvalues are at most $d$. Note that under the $z$-coordinate system, $P_i$ is given by

$$U P_i U^\mathsf{T} = \begin{bmatrix} 0_{m \times m} & \\ & I_{d-m} \end{bmatrix} \text{ since } D_i = \begin{bmatrix} [\geq d]_{m \times m} & \\ & [\leq d]_{(d-m) \times (d-m)} \end{bmatrix}.$$

Hence, under the $z$-coordinate system, the two ellipsoids above can be written as

$$(I + P_i)^2 \to (I + U P_i U^\mathsf{T})^2 = \begin{bmatrix} I_m & \\ & 2I_{d-m} \end{bmatrix}^2, \tag{B.3}$$

$$\Sigma_{i+1} \to U\Sigma_{i+1}U^\mathsf{T} = U(I + P_i)U^\mathsf{T} \cdot U\Sigma_i U^\mathsf{T} \cdot U(I + P_i)U^\mathsf{T}$$

$$= \underbrace{\begin{bmatrix} I_m & \\ & 2I_{d-m} \end{bmatrix}}_{=: \overline{D}} U\Sigma_i U^\mathsf{T} \begin{bmatrix} I_m & \\ & 2I_{d-m} \end{bmatrix}. \tag{B.4}$$

Let $c_\mathcal{A}$ and $c_\mathcal{B}$ denote the centers of $\mathcal{A}$-ellipsoid and $\mathcal{B}$-ellipsoid in the $z$-coordinate system. We define $\mathcal{S} = \text{span}(\{e_{m+1}, \ldots, e_d\})$ (i.e. the span of the deficient axes) and $\mathcal{S}^\perp = \text{span}(\{e_1, \ldots, e_m\})$ (i.e. the span of the sufficient axes).

**Case I:** $0 < m < d$**.** We first show that in the $z$-coordinate system, $\mathcal{K}_{i+1}$ contains two lower-dimensional balls

$$\mathsf{B}_\mathcal{S} := B_{2\lambda r_i}(c_\lambda) \cap (\mathcal{S} + c_\lambda) \quad \& \quad \mathsf{B}_{\mathcal{S}^\perp} := B_{\frac{9(1-\lambda)}{10} d^{1/2}}(c_\lambda) \cap (\mathcal{S}^\perp + c_\lambda),$$

where $c_\lambda := \lambda c_\mathcal{A} + (1 - \lambda)c_\mathcal{B}$ and $\lambda := (1 - 1/\log d)^{1/2}$.

By (B.3), the part of the $\mathcal{A}$-ellipsoid along $\mathcal{S}$ (i.e., $\mathcal{A} \cap (\mathcal{S} + c_\mathcal{A})$) contains $B_{2r_i}(c_\mathcal{A})$. By convexity of $\mathcal{K}_{i+1}$, it contains $B_{2\lambda r_i}(c_\lambda)$ along the affine subspace $\mathcal{S} + c_\lambda$, since $\mathcal{K}_{i+1}$ contains $c_\mathcal{B}$. On the good covariance-estimation event, $\widehat{\Sigma}_i \preceq \frac{11}{10}\Sigma_i + \frac{d}{100} I_d$. Conjugating by $U^\mathsf{T}\overline{D}$, this is equivalent to

$$\overline{D}D_i\overline{D} \preceq \frac{11}{10}\overline{D}U\Sigma_i U^\mathsf{T}\overline{D} + \frac{d}{100}\overline{D}^2 \underset{(B.4)}{=} \frac{11}{10}U\Sigma_{i+1}U^\mathsf{T} + \frac{d}{100}\overline{D}^2 .$$

Then the projection onto $\mathcal{S}^\perp$ (i.e., taking the top-left $m \times m$ block matrix) results in

$$(U\Sigma_{i+1}U^\mathsf{T})|_{\mathcal{S}^\perp} \succeq \frac{9}{10}dI_m ,$$

so the $\mathcal{B}$-ellipsoid along $\mathcal{S}^{\perp}$ contains $B_{0.9d^{1/2}}(c_{\mathcal{B}})$. Since the $\mathcal{A}$-ellipsoid along $\mathcal{S}^{\perp}$ contains $B_{r_i}(c_{\mathcal{A}})$ as seen in (B.3), the convexity of $\mathcal{K}_{i+1}$ implies that it contains a ball centered at $c_{\lambda}$ of radius $\lambda r_i + (1 - \lambda) \cdot \frac{9}{10}d^{1/2} \geq \frac{9(1-\lambda)}{10}d^{1/2}$.

We now prove that $\mathcal{K}_{i+1}$ contains a ball of radius roughly $2r_i$ centered at $c_{\lambda}$, by showing that any point in such a ball can be written as a convex combination of $\mathsf{B}_{\mathcal{S}}$ and $\mathsf{B}_{\mathcal{S}^{\perp}}$. More specifically, we may assume $c_{\lambda} = 0$ by translation and denote by $P_{\mathcal{S}}$ the orthogonal projection onto $\mathcal{S}$. For any $x$ with $\|x\| \leq 2\gamma\lambda r_i$ and $\gamma$ to be determined, we write

$$x = (1-t)\frac{(I - P_{\mathcal{S}})x}{1-t} + t\frac{P_{\mathcal{S}}x}{t} \quad \text{for } t \in (0,1) \, .$$

For $u := (I - P_{\mathcal{S}})x$ and $v := P_{\mathcal{S}}x$, we will show that if $\|u\|^2 + \|v\|^2 = \|x\|^2 \leq 4\gamma^2\lambda^2 r_i^2$, then

$$\exists t \in (0,1) : \quad \frac{\|u\|}{1-t} \leq \frac{9(1-\lambda)}{10}d^{1/2} \quad \text{and} \quad \frac{\|v\|}{t} \leq 2\lambda r_i$$

$$\iff \quad \exists t \in (0,1) : \frac{\|v\|}{2\lambda r_i} \leq t \leq 1 - \frac{10\|u\|}{9(1-\lambda)d^{1/2}}$$

$$\iff \quad \frac{10\|u\|}{9(1-\lambda)d^{1/2}} + \frac{\|v\|}{2\lambda r_i} = \frac{20\lambda r_i}{9(1-\lambda)d^{1/2}}\,a + b \leq 1 \quad \text{for } a := \frac{\|u\|}{2\lambda r_i} \text{ and } b := \frac{\|v\|}{2\lambda r_i} \, .$$

To this end, it suffices to show that the maximum of the following problem is at most 1:

$$\max \frac{20\lambda r_i}{9(1-\lambda)d^{1/2}}\,a + b \quad \text{subject to } a^2 + b^2 = \gamma^2 \, .$$

Using the Lagrange multiplier method, the maximum is attained if $(a,b) = \lambda(\frac{20\lambda r_i}{9(1-\lambda)d^{1/2}}, 1)$ for some $\lambda > 0$. Solving $a^2 + b^2 = \gamma^2$ with this condition, we have $\lambda = \gamma(1 + \frac{400\lambda^2 r_i^2}{81(1-\lambda)^2 d})^{-1/2}$. Thus, the maximum is

$$\frac{20\lambda r_i}{9(1-\lambda)d^{1/2}}\,a + b = \gamma\left(1 + \frac{400\lambda^2 r_i^2}{81(1-\lambda)^2 d}\right)^{1/2},$$

which can be bounded by 1 by setting $\gamma = (1 + \frac{400\lambda^2 r_i^2}{81(1-\lambda)^2 d})^{-1/2}$.

We now show that $\gamma^{-2} \leq \lambda^{-2}$ (so $\gamma \geq \lambda$). Using $2^{10}r_i^2 \log^4 d \leq d$ and $\lambda \leq 1 - 1/2\log d \leq 1$, we have

$$\gamma^{-2} = 1 + \frac{400\lambda^2 r_i^2}{81(1-\lambda)^2 d} \leq 1 + \frac{400\lambda^2}{81 \cdot 2^{10}(1-\lambda)^2 \log^4 d} \leq 1 + \frac{1}{12\log^2 d} \, .$$

As $\lambda^{-2} = 1 + \frac{1}{\log d - 1}$, we clearly have $\gamma^{-2} \leq \lambda^{-2}$. Therefore,

$$\mathrm{inrad}(\mathcal{K}_{i+1}) \geq 2\gamma\lambda r_i \geq 2\lambda^2 r_i \, .$$

**Case II: $m = 0$ or $d$.** When $m = 0$, the $\mathcal{A}$-ellipsoid is simply the $d$-dimensional $B_{2r_i}(c_{\mathcal{A}})$. If $m = d$, then it means that the $\mathcal{B}$-ellipsoid contains the $d$-dimensional ball $B_{\frac{9}{10}d^{1/2}}(c_{\mathcal{B}})$. Since $2^{10}r_i^2 \log^4 d \leq d$, we have $\frac{9}{10}d^{1/2} \geq 2\lambda^2 r_i$, where $\lambda$ is as before. Combining these two cases justifies $\mathrm{inrad}(\mathcal{K}_{i+1}) \geq 2\lambda^2 r_i$.

Lastly, if the two centers $c_i$ and $\mu_i$ are the same, then $c_{\mathcal{A}} = c_{\mathcal{B}}$. In such case, the same argument goes through (with an even larger radius). $\qquad\square$

**Final guarantee.**   Putting these together, we can prove our main result in this section.

*Proof of Lemma 4.6.* We would like to exclude three bad events — (1) failure of Gaussian cooling in Line 2, (2) failure of In-and-Out throughout the algorithm, and (3) failure of covariance estimation in Line 5 within each while-loop and in Line 8.

As for (1), we can simply set the failure of the warm-start generation to $\mathbb{P}/d$ with logarithmic overhead in $d$ (Kook & Zhang, 2025). As for (2), we already picked $h_i$ and $N_i$ in Line 4 so that the total failure probability of In-and-Out is at most

$\mathfrak{p}/d$ (see the proof of Lemma 4.2). As for (3), by Lemma 4.4, the failure probability throughout the while-loop is at most $\mathfrak{p} \log d/d$ by the union bound. As in Lemma 4.3, the current warm point is $O(1)$-warm with respect to the terminal target, so Lemma A.1 only changes the failure probability by a universal factor. As for the final covariance estimation in Line 8, by the multiplicative form of covariance estimation guarantees (Corollary 3.6), there exists a universal constant $c' > 0$ such that at termination, writing $i_\star = i$ and $\Sigma_\star := \Sigma_{i_\star}$, $|\widehat{\Sigma} - \Sigma_\star| \preceq \frac{1}{10} \Sigma_\star$ with probability at least $1 - \frac{\mathfrak{p}}{d}$ when the final retained sample size is at least $c' d \log^6 d$.

Putting these estimates together, the algorithm succeeds with probability at least $1 - \mathfrak{p} \log d/d - \mathfrak{p}/d \geq 1 - \mathfrak{p}/\sqrt{d}$, ensuring that the covariance of the transformed convex body satisfies

$$\frac{1}{1.1} I_d \preceq \widehat{\Sigma}^{-1/2} \Sigma_\star \widehat{\Sigma}^{-1/2} \preceq \frac{10}{9} I_d \,.$$

Therefore, $\widehat{\Sigma}^{-1/2}(\mathcal{K}_{i_\star} - \widehat{\mu}) = \widehat{\Sigma}^{-1/2}(T_{i_\star}\mathcal{K} - \widehat{\mu})$ is 2-isotropic.

As for the query complexity, Algorithm 1 starts with Gaussian cooling for the warm-start generation, using $\widetilde{O}(C^2 d^3)$ membership queries. In each while-loop, since In-and-Out from an $O(1)$-warm start iterates $k_i N_i = \widetilde{O}(cC^4 d^3) = \widetilde{O}(C^4 d^3)$ times, it uses $\widetilde{O}(C^4 d^3)$ membership queries. Since the number of while-loops is at most $2 \log d$, the total query complexity of the while-loops is $\widetilde{O}(C^4 d^3)$. The final estimation step uses $c' d \log^6 d$ calls to In-and-Out$_N$, each with $N = \widetilde{O}(C^2 d^2)$, hence costs $\widetilde{O}(C^2 d^3)$, which is dominated by $\widetilde{O}(C^4 d^3)$. $\square$

*Remark* B.2 (General bodies). We can design a rounding algorithm with roughly $d^{3.5}$ query complexity for a *general convex body* containing a unit ball, which combines Algorithm 1 with the annealing algorithm in Jia et al. (2021). Given a well-rounded uniform distribution over $T(\mathcal{K} \cap B_r(0))$ for some $r \geq 1$ and affine map $T : \mathbb{R}^d \to \mathbb{R}^d$, Algorithm 1 will find a new affine map $T'$ such that $T'(\mathcal{K} \cap B_r(0))$ is nearly isotropic with high probability. The annealing algorithm then moves to the uniform distribution over $T'(\mathcal{K} \cap B_{r(1+d^{-1/2})}(0))$, which can be shown to be well-rounded by Jia et al. (2024, Lemma 3.4). This annealing algorithm begins with a ball of radius $r = 1$ (which is surely well-rounded) and repeats the procedure above until the radius reaches the diameter $D$ of $\mathcal{K}$. The total number of iterations for this annealing procedure is $d^{1/2} \log D$. Multiplying by the complexity of Algorithm 1, we find that the total complexity of the entire rounding algorithm is $\widetilde{O}(d^{3.5} \log D)$.

### B.3. Covariance estimation for unconstrained distributions

The following ensures that we can tractably initialize our algorithm not too far from the target.

**Lemma B.3** (Initialization, Chewi et al. (2022, Lemma 32)). *Suppose $\nabla V(0) = 0$ and $V$ is $\beta$-smooth (Assumption 4.7). Let $\pi \propto \exp(-V)$ and $\mathfrak{m} = \mathbb{E}_\pi \|\cdot\|$. Then, initialization $\mu_0 = \mathcal{N}(0, \frac{1}{2\beta} I_d)$ satisfies*

$$\log \chi^2(\mu_0 \,\|\, \pi) \lesssim 1 + \beta + V(0) - \min V + d \log(\mathfrak{m}^2 \beta) \,.$$

The assumption that $\nabla V(0) = 0$ is standard, as a local extremum can be found by optimization algorithms, which have complexity dominated by that of the sampling procedure. Assuming mild values for all parameters of interest, this gives a bound of $\widetilde{O}(d \vee \beta)$.

*Proof of Lemma 4.11.* Here, we elucidate a few details when adapting the results of Altschuler & Chewi (2024). The following steps are already present in their analysis of the proximal sampler, but we reemphasize the derivation here for full clarity. Namely, by the analysis of Altschuler & Chewi (2024, Appendix D.4), the condition number of the backwards part of the proximal sampler is $\Theta(1)$. Furthermore, since the forward part of the proximal sampler is implemented exactly, then denoting $P_{\text{forward}}, P_{\text{backward}}$ for the two exact kernels with step size $h \asymp 1/\beta$, and $\hat{P}_{\text{backward}}$ for the approximate kernel from Altschuler & Chewi (2024),

$$\mathsf{KL}(\delta_x \hat{P} \,\|\, \delta_x P) \leq \mathsf{KL}(\delta_x P_{\text{forward}} \,\|\, \delta_x P_{\text{forward}}) + \mathbb{E}_{y \sim \delta_x P_{\text{forward}}} \mathsf{KL}(\delta_y \hat{P}_{\text{backward}} \,\|\, \delta_y P_{\text{backward}}) \leq \varepsilon \,,$$

using Altschuler & Chewi (2024, Theorem D.1) as claimed. $\square$

*Proof of Lemma 4.13.* For any $z_1 \in \mathbb{R}^d$ and $m \in \mathbb{N}$, using the data-processing inequality and then the chain rule for KL,

$$\mathsf{KL}(\delta_{z_1} \hat{P}^m \,\|\, \delta_{z_1} P^m) \leq \mathsf{KL}(\delta_{z_1} \hat{P} \,\|\, \delta_{z_1} P) + \mathbb{E}_{z_2 \sim \delta_{z_1} \hat{P}} \mathsf{KL}(\delta_{z_2} \hat{P} \,\|\, \delta_{z_2} \hat{P})$$

$$+ \cdots + \mathbb{E}_{z_m \sim \delta_{z_1} \hat{P}^{m-1}} \mathsf{KL}(\delta_{z_m} \hat{P} \,\|\, \delta_{z_m} \hat{P})\,.$$

This heavily uses the fact that the joint distribution arises from a Markov chain.

The query complexity to bound the left side by $\varsigma \in (0, 1/2]$ is $\widetilde{O}(md^{1/2} \log^3 \frac{1}{\varsigma})$, obtained by choosing an error of $\varsigma/m$ for the kernel and then applying Lemma 4.11 $m$-times. Then,

$$
\begin{aligned}
2\,\|\nu_{1:K} - \hat{\nu}_{1:K}\|_{\mathsf{TV}}^2 &\le \mathsf{KL}(\hat{\nu}_{1:K} \,\|\, \nu_{1:K}) \\
&\le \mathsf{KL}(\hat{\nu} \,\|\, \nu) + \mathbb{E}_{x_1 \sim \hat{\nu}} \mathsf{KL}(\delta_{x_1} \hat{P}^n \,\|\, \delta_{x_1} P^n) \\
&\quad + \cdots + \mathbb{E}_{x_{K-1} \sim \hat{\nu} \hat{P}^{(K-1)n}} \mathsf{KL}(\delta_{x_{K-1}} \hat{P}^n \,\|\, \delta_{x_{K-1}} P^n)\,.
\end{aligned}
$$

Therefore, the query complexity of bounding the left side by $2\delta^2$ is $\widetilde{O}((Kn + n_0)d^{1/2} \log^3 \frac{Kn+n_0}{\delta})$. $\qquad\square$

*Proof of Theorem 4.14 and Remark 4.15.* For the choices of $h \asymp 1/\beta$, $n = O(\kappa)$ given, the spectral gap of $P^n$ will be bounded below by 0.99. Thus, choosing $K = \widetilde{\Theta}(\frac{d\phi}{\varepsilon^2})$, the covariance estimator using samples from the exact kernel $P^n$ will concentrate around $\Sigma$. Precisely, if

$$\overline{\Sigma} = \frac{1}{K} \sum_{j=1}^{K} \Big(Y_j - \frac{1}{K} \sum_{k=1}^{K} Y_k\Big)^{\otimes 2} =: F(Y_1, \ldots, Y_K)\,,$$

where $Y_1 \sim \nu$ and $Y_{i+1} \sim \delta_{Y_i} P^n$, then Theorem 3.5 combined with Lemma A.1 tell us for our choice of parameters that with probability $1 - \sqrt{\mathfrak{p}/d}$,

$$\|\overline{\Sigma} - \Sigma\| \le \varepsilon \|\Sigma\|\,.$$

Here, we use that our initial distribution $\nu = \mu_0 P^{n_0}$ comes from applying $P$ enough iterations that $\chi^2(\nu \,\|\, \pi) \lesssim 1$, and so via Lemma B.3 our choice of $n_0$ is sufficient. Note that we can also write $\hat{\Sigma} = F(X_1, \ldots, X_k)$, where $\{X_j\}_{j \le K}$ is generated using $\hat{P}^n$ instead.

Denote the undesirable event on $\mathbb{R}^{K \times d}$ by

$$\mathcal{A} := \big\{(X_1, \ldots, X_K) \in \mathbb{R}^{d \times K} : \|F(X_1, \ldots, X_K) - \Sigma\| > \varepsilon \|\Sigma\|\big\}\,.$$

Under the approximate kernel $\hat{P}^n$, we have via Lemma 4.13 and the definition of the TV distance that

$$\hat{\nu}_{1:K}(\mathcal{A}) \le \nu_{1:K}(\mathcal{A}) + |\nu_{1:K}(\mathcal{A}) - \hat{\nu}_{1:K}(\mathcal{A})| \le \nu_{1:K}(\mathcal{A}) + \|\nu_{1:K} - \hat{\nu}_{1:K}\|_{\mathsf{TV}}\,.$$

Choosing $\delta \asymp \mathfrak{p}/\sqrt{d}$ in Lemma 4.13 to make this TV distance bounded by $\mathfrak{p}/\sqrt{d}$, and adding it to our bound under $\nu_{1:K}$ from earlier, we obtain a bound of $1 - \mathfrak{p}/\sqrt{d}$ on the probability of failing to estimate the covariance under the approximate implementation of the kernel.

The total expected query complexity is then given by

$$N \le \widetilde{O}\big(K\kappa d^{1/2} \log^3(K + n_0)\big) + \widetilde{O}\big(n_0 d^{1/2} \log^3(K + n_0)\big)\,,$$

where the second term is the cost of generating the initial iterate, and the first term comes from the generation of the remaining iterates. Putting all these together concludes the proof.

For the bound given in Remark 4.15, we note that we will again need $K = \widetilde{\Theta}(\frac{d\phi}{\varepsilon^2})$ iterates in total. By the same principle as Lemma 4.13, it suffices that each iterate is $O(\frac{1}{Kd})$ close to the target in $\sqrt{\mathsf{KL}}$, and we should again choose $\varrho$ identically to the Markovian case. To produce a single iterate, we require $\widetilde{O}(n_0 d^{1/2} \operatorname{polylog}(K + n_0))$ queries in expectation. Putting these all together concludes the proof. $\qquad\square$

## C. Finer bound on the tail

**Lemma C.1.** *Under the same assumptions as Lemma A.7, we in fact have the improved bound*

$$\mathbb{P}\Big(\frac{1}{N} \sum_i \|X_i\|^2 \mathbb{1}_{B^c} - \mathbb{E}\Big[\frac{1}{N} \sum_i \|X_i\|^2 \mathbb{1}_{B^c}\Big] \ge s\rho\Big) \le \frac{1}{s^2}\,.$$

*Here, $\rho$ can be bounded as $\rho^2 \le \frac{1}{N}(1 + \frac{2}{\lambda})(\operatorname{tr}\Sigma + C_{\mathsf{PI}}(\pi))^2 e^{-t/2}$.*

*Proof.* Note that

$$\left\| \frac{1}{N} \sum X_i^{\otimes 2} \mathbb{1}_{B^c} \right\| \leq \frac{1}{N} \sum \sup_v |X_i \cdot v|^2 \mathbb{1}_{B^c} = \frac{1}{N} \sum_i \|X_i\|^2 \mathbb{1}_{B^c}.$$

Consider the centered function $f(X_i) := \{\|X_i\|^2 \mathbb{1}_{B^c} - \mathbb{E}\|X_i\|^2 \mathbb{1}_{B^c}\}$, in which case we can write

$$\mathbb{E}\Big[\Big\{ \frac{1}{N} \sum_{i=1}^N \|X_i\|^2 \mathbb{1}_{B^c} - \mathbb{E}\Big[ \frac{1}{N} \sum_{i=1}^N \|X_i\|^2 \mathbb{1}_{B^c} \Big] \Big\}^2 \Big]$$

$$= \frac{1}{N^2} \mathbb{E}[(\sum_{i=1}^N f(X_i))^2] = \frac{1}{N^2} \sum_{i=1}^N \mathbb{E}[(f(X_i))^2] + \frac{2}{N^2} \sum_{i=1}^N \mathbb{E}[\sum_{\substack{j>i \\ j \in [N]}} f(X_i)f(X_j)]$$

$$= \frac{1}{N} \mathbb{E}[(f(X_i))^2] + \frac{2}{N^2} \sum_{i=1}^N \mathbb{E}[f(X_i) \sum_{\substack{j>i \\ j \in [N]}} \mathbb{E}[f(X_j)|X_i]]$$

$$= \frac{1}{N} \mathbb{E}[(f(X_i))^2] + \frac{2}{N^2} \sum_{i=1}^N \mathbb{E}[f(X_i) \sum_{\substack{k>0 \\ k \in [N-i]}} P^k f(X_i)].$$

Consider just the second term. If we observe just the even summands, we see that they can be bounded as

$$\mathbb{E}\Big[ f(X_i) \sum_{\substack{k>0 \\ k \in [N-i]}} P^{2k} f(X_i) \Big] = \sum_{\substack{k>0 \\ k \in [\lfloor N-i \rfloor]}} \mathbb{E}[|P^k f|^2] \leq \sum_{k>0} \lambda^k \mathbb{E}[|f|^2] = \frac{1}{1-\lambda} \mathbb{E}[|f|^2],$$

The odd terms can be handled via Cauchy–Schwarz, $\mathbb{E}[fP^{k+1}f] \leq \sqrt{\mathbb{E}f^2 \mathbb{E}[(P^{2k+2}f)^2]} \leq \lambda^{k+1} \mathbb{E}[f^2]$. Thus, the effect is to approximately double the bound above.

Note that $\mathbb{E}[(f(X_i))^2]$ can be bounded by

$$\mathbb{E}[\|X_i\|^4 \mathbb{1}_{B^c}] = \mathbb{E}[\|X\|^4 \mathbb{1}_{B^c}] \leq \sqrt{\mathbb{E}[\|X\|^8]} \sqrt{\mathbb{P}(\|X\| \geq t\sqrt{\operatorname{tr}\Sigma})}$$

$$\lesssim \big(\operatorname{tr}\Sigma + C_{\mathsf{PI}}(\pi)\big)^2 \cdot e^{-t/2},$$

since

$$\mathbb{P}(\|X\| - \mathbb{E}\|X\| > t) \lesssim \exp\big(-\frac{t}{\sqrt{C_{\mathsf{PI}}(\pi)}}\big),$$

and integrating, bounding $\mathbb{E}\|X\| \lesssim \sqrt{\operatorname{tr}\Sigma}$. The mean can be bounded by

$$\mathbb{E}\Big[ \frac{1}{N} \sum_i \|X_i\|^2 \mathbb{1}_{B^c} \Big] = \mathbb{E}[\|X\|^2 \mathbb{1}_{B^c}] \leq \sqrt{\mathbb{E}[\|X\|^4]} \sqrt{\mathbb{P}(\|X\| \geq t\sqrt{\operatorname{tr}\Sigma})} \lesssim \operatorname{tr}\Sigma \cdot e^{-t/2}.$$

This implies that

$$\rho^2 := \operatorname{var}\Big( \frac{1}{N} \sum_{i=1}^N f \Big) \leq \frac{1}{N}\Big(1 + \frac{2}{\lambda}\Big)\big(\operatorname{tr}\Sigma + C_{\mathsf{PI}}(\pi)\big)^2 e^{-t/2}.$$

Recall that $\lambda$ is an absolute constant which is bounded away from zero.

Finally, by Chebyshev's inequality,

$$\mathbb{P}\Big( \frac{1}{N} \sum_i \|X_i\|^2 \mathbb{1}_{B^c} - \mathbb{E}\Big[ \frac{1}{N} \sum_i \|X_i\|^2 \mathbb{1}_{B^c} \Big] \geq s\rho \Big) \leq \frac{1}{s^2},$$

which completes the proof. $\qquad\square$

