# OpenReview forum: "Covariance estimation using Markov chain Monte Carlo"
_ICML.cc/2026/Conference — ICML 2026 regular_

### Official Review · Reviewer_voTy · 2026-03-03

**Soundness:** 4
**Presentation:** 3
**Significance:** 3
**Originality:** 3
**Overall Recommendation:** 3
**Confidence:** 1

**Summary:**

The paper deals with the properties and applications of the sample mean and sample covariance using non-iid samples.

**Compliance With Llm Reviewing Policy:**

Affirmed.

**Key Questions For Authors:**

Please point out a few papers and applications from recent ICML or NeurIPS venues where the contributions can make an impact. Can you demonstrate this with some experiments following up on these papers?

**Limitations:**

yes

**Strengths And Weaknesses:**

First a disclaimer - i know covariance estimation but this paper is very far from me. The problem is probably on my side, but I found it hard to appreciate its contributions. If I understood correctly, then this is a follow up on Jia's work from STOC and was also used in Kook from STOC. I therefore think STOC is a better venue for this kind of work which may be very strong but less suitable for ICML.

At the minimum, I would expect some experiments and applications closer to modern machine learning and the ICML community.

---

> ### Author Rebuttal · Authors · 2026-03-30
>
> We would like to clarify the main contributions of this work, which we believe are directly relevant to the ICML community.
>
> The central contribution is not merely an improvement of specific STOC results, but rather the development of a *general framework* for mean and covariance estimation from dependent MCMC samples. Concretely, we identify the spectral gap of the Markov chain and the Poincaré constant of the target distribution as a *promising* pair of assumptions for controlling statistical estimation error under sample dependence, and use them to derive theoretical bounds on both sample and query complexity. The application to Jia et al. (STOC'21) [0] is presented as a concrete application of this framework---one that yields a simpler and more principled proof---but the framework itself applies broadly as seen in the covariance estimation of unconstrained distributions (Section 4.2). Furthermore, we highlight that unlike a standard STOC submission (which primarily emphasizes computational complexities), our result contains substantial statistical components (concentration of various estimators) that makes it better suited for an ML venue.
>
> We also highlight a practical motivation that is directly relevant to machine learning practitioners. A standard workflow in applied Bayesian statistics and MCMC-based inference is to *run a single Markov chain and collect samples every $N_{\mathrm{mix}}$ steps* for downstream statistical tasks such as posterior mean and covariance estimation. Despite being near-universal in practice, the theoretical justification for this procedure---specifically, that the resulting correlated samples suffice for accurate covariance estimation---was not previously established under our pair of assumptions. Our framework provides exactly this justification, giving practitioners a principled guarantee for a procedure they already use.
>
> Finally, the field of Monte Carlo methods is quite active within ICML and NeurIPS. We point out the papers [1, 2] from last year's NeurIPS, and [3, 4, 5] from last year's ICML. We also mention the spotlighted paper [6] from NeurIPS 2024, which contains material directly relevant to the current work (particularly §4.1). We note that most of these works did not contain an experimental component. We hope the reviewer will reconsider their score.
>
> **References**
>
> **[0]** Jia, H., Laddha, A., Lee, Y. T., & Vempala, S. (2021, June). Reducing isotropy and volume to kls: an o*(n 3 ψ 2) volume algorithm. In Proceedings of the 53rd Annual ACM SIGACT Symposium on Theory of Computing (pp. 961-974).
>
> **[1]** Renaud, M., et al. "From stability of Langevin diffusion to convergence of proximal MCMC for non-log-concave sampling." *arXiv:2505.14177* (2025).
>
> **[2]** Guan, Y., Balasubramanian, K., and Ma, S. "Riemannian proximal sampler for high-accuracy sampling on manifolds." *arXiv:2502.07265* (2025).
>
> **[3]** Yang, B., and Wang, X. "Non-asymptotic Error Bounds in $\mathcal{W}_2$-Distance with Sqrt(d) Dimension Dependence and First Order Convergence for Langevin Monte Carlo beyond Log-Concavity." *ICML 2025*.
>
> **[4]** Feng, S., and Yang, Y. "Regularized Langevin dynamics for combinatorial optimization." *ICML 2025*.
>
> **[5]** Hu, J., Ma, Y.-T., and Eun, D. Y. "Beyond Self-Repellent Kernels: History-Driven Target Towards Efficient Nonlinear MCMC on General Graphs." *ICML 2025*.
>
> **[6]** Kook, Y., Vempala, S. S., and Zhang, M. S. "In-and-out: Algorithmic diffusion for sampling convex bodies." *NeurIPS 2024* (Spotlight).

---

> > ### Author Rebuttal · Reviewer_voTy · 2026-04-01
> >
> > Thank you for the rebuttal. It is clear that this a matter of taste and I guess I am just more interested in applied works. I appreciate the importance of theoretical works too and therefore specifically chose a low confidence. I will let the AC decide based on the more theoretical reviewers. Good luck

---

### Official Review · Reviewer_VoRM · 2026-03-04

**Soundness:** 3
**Presentation:** 3
**Significance:** 3
**Originality:** 3
**Overall Recommendation:** 5
**Confidence:** 4

**Summary:**

This paper studies the sample complexity of covariance estimation problem given samples generated from a Markov chain.

They show both additive and multiplicative concentration results for classic MLE covariance estimator if the distribution satisfies PI and the Markov chain has constant spectral gap.

Based on this framework, the authors give two applications:

(1) They design a method that solves the isotropic rounding problem with query complexity $\tilde{O} (d ^ 3)$ , while best prior work requires $\tilde{O}(d ^ {3.5})$ queries.

(2) For Gibbs measures with PI constant $\alpha^{-1}$ and $\beta$-smooth energy, they give an covariance estimator that is $\epsilon$-accurate with query complexity $\tilde{O}(\kappa d ^ {3/2} / \epsilon^2)$, which is strictly better than current best known $\tilde{O}(\kappa d ^ {5/2} / \epsilon^2)$ result for i.i.d. samples.

**Compliance With Llm Reviewing Policy:**

Affirmed.

**Final Justification:**

Soundness:
This paper is technically sound, every statement is justified rigorously and properly.

Originality & Significance:
This paper studies the sample complexity of covariance estimation problem given samples generated from a Markov chain. The problem itself is novel and interesting. The authors provide two applications of their proposed framework: (a) faster isotropic rounding procedure of convex bodies; (b) faster covariance estimation of Gibbs measure, indicating the significance of their theoretical results.

Clarity:
The paper is well written and easy to follow. However, their discussion of query and sample complexity may be misleading to non-expert readers, which they promised to modify in the revised version.

As other reviewers suggest, this paper mainly establishes theoretical results closely related to TCS and their relations to ML field is unclear. I have different opinions on this due to two reasons:
1. The framework proposed by this paper can be viewed as a sub-problem of learning from non i.i.d. samples, which deserves more attention due to the wide use of synthetic data recently. Assuming the samples are generated from a fast-mixing MC is a reasonable and interesting way to model  "non-i.i.d. samples" and may be of independent interest.

2. The case study section all lies in classic settings of sampling theory, which is a core topic of ML but may be strange to non-expert readers.

Therefore, the reviewer recommend acceptance.

**Key Questions For Authors:**

Key questions:
1. Do we have a natural analogue result for the discrete space? If the extension is not direct, can the authors briefly explain the technical hardness?
2. Is it possible to extend the PI assumption w.r.t. $\lVert \nabla f\rVert^2$ to the generalized Dirichlet form? It directly affects the tail behavior, but could the authors give more detailed comments on that?
3. Under Assumption 3.1, the author mentioned a subtle cost if we can't choose $\pi_0 = \pi$. Can the author explain how the handle this cost for the isotropic rounding algorithm? Since $X_0$ is not uniformly from the convex body, this cost should show up in the analysis.

Comments on potentially (minor) issues:
1. The authors did not give a formal definition of query complexity until Section 4. Given the fact that the query complexity is only discussed in the case study section and the query oracle is used to construct and run MCMC, maybe it is better to just introduce sample complexity in the covariance estimation task and discuss the query complexity only in the case study section.
2.  Please consider giving the term "well-rounded" a formal definition. In particular, on line 56, should the reader view $\mathbb{E} _ \pi [\lVert \cdot \rVert ^ 2] \lesssim d$ as the interpretation of "well-rounded" or another assumption?

**Limitations:**

Yes

**Strengths And Weaknesses:**

Strength:
1. Learning from non-i.i.d. samples is an interesting problem that may have potential applications for many downstream areas. In particular, modeling samples as Markovian iterates is not only novel but also realistic for many real-world scenarios.
2. The paper is mathematically solid. No major mistake is identified.
3. Their case studies are both well-known and important problems in ML field. Getting improved results in these settings hugely strengthens the applicability and potential impact of their proposed theoretical framework.

Weakness:
The writing and presentation could be improved by giving clearer statements and more detailed explanations for certain terms and settings that are commonly used in TCS and learning theory fields but may be new to non-expert readers and general audience.

---

> ### Author Rebuttal · Authors · 2026-03-30
>
> > **Q1.** Do we have a natural analogue result for the discrete space? If the extension is not direct, can the authors briefly explain the technical hardness?
>
> A discrete-space analogue is possible in principle. The Poincaré inequality has a natural discrete counterpart, and matrix Bernstein inequalities under Markovian assumptions apply in discrete space as well [1]. We believe the extension is feasible but non-trivial and leave it for future work.
>
> > **Q2.** Is it possible to extend the PI assumption w.r.t. $\|\nabla f\|^2$ to the generalized Dirichlet form? It directly affects the tail behavior, but could the authors give more detailed comments on that?
>
> The PI in our work uses the standard Dirichlet form $\mathcal{E}(f,f) = \mathbb{E}_\pi[\|\nabla f\|^2]$. For a generalized Dirichlet form $\mathcal{E}(f,f) = \int \Gamma(f,f)\,d\pi$ with carré du champ operator $\Gamma$, the Lipschitz concentration (2.1) would need to be adapted to reflect the geometry induced by $\Gamma$. The tail behavior would then depend on the operator norm of $\Gamma$ rather than the Euclidean gradient. We believe the main results carry over with appropriate modifications to the Lipschitz constant.
>
> > **Q3.** Under Assumption 3.1, the author mentioned a subtle cost if we can't choose $\pi_0 = \pi$. Can the author explain how to handle this cost for the isotropic rounding algorithm? Since it is not uniformly from the convex body, this cost should show up in the analysis.
>
> As stated after Assumption 3.1, the cost of starting from $\pi_0 \neq \pi$ introduces a multiplicative factor of $1 + \chi^2(\pi_0 \| \pi)$ in the failure probabilities, via Lemma A.1. For the isotropic rounding algorithm (Section 4.1), this is handled in Lemma 4.3: the warm start generated by Gaussian cooling satisfies $R_\infty(\nu \| \pi_{K_1}) \leq \log 2$, which implies $\chi^2(\nu \| \pi_{K_1}) \leq 1$. Hence, the additional factor is at most $2$, which is absorbed into the (multiplicative) universal constant in failure probabilities and the number of samples (and therefore the number of queries).
>
> > **C1.** The authors did not give a formal definition of query complexity until Section 4. Given the fact that the query complexity is only discussed in the case study section and the query oracle is used to construct and run MCMC, maybe it is better to just introduce sample complexity in the covariance estimation task and discuss the query complexity only in the case study section.
>
> We agree and will move the formal definition of query complexity to Section 1 (or Section 2), with a note that it applies only to the application sections.
>
> > **C2.** Please consider giving the term "well-rounded" a formal definition. In particular, on line 56, should the reader view $\mathbb{E}_\pi[\|\cdot\|^2] \lesssim d$ as the interpretation of "well-rounded" or another assumption?
>
> We agree with the reviewer; well-roundedness is the same as $\mathbb{E}_\pi[\|\cdot\|^2] \lesssim d$, and we will make this explicit in the revision.
>
> **References**
>
> **[1]** Garg, A., Lee, Y. T., Song, Z., and Srivastava, N. A matrix expander Chernoff bound. In Symposium on Theory of Computing (STOC), pp. 1102–1114, 2018.

---

> > ### Author Rebuttal · Reviewer_VoRM · 2026-03-31
> >
> > Thanks for the detailed response. My questions are well addressed and I shall increase my confidence level.

---

### Official Review · Reviewer_oeXH · 2026-03-06

**Soundness:** 3
**Presentation:** 3
**Significance:** 3
**Originality:** 3
**Overall Recommendation:** 5
**Confidence:** 2

**Summary:**

This paper presents the first systematic study of covariance estimation using MCMC-dependent samples under the assumptions that the target distribution satisfies a Poincaré inequality and the Markov chain has a spectral gap. The authors derive concentration inequalities for the sample covariance matrix and provide explicit sample complexity bounds. The framework is applied to two important problems: 1) isotropic rounding of convex bodies; 2) covariance estimation for unconstrained distributions.

**Compliance With Llm Reviewing Policy:**

Affirmed.

**Key Questions For Authors:**

1.How can the Poincaré inequality and spectral gap λ ≥ 0.99 be verified in practice? Does the method apply to distributions that do not satisfy these assumptions, such as multimodal distributions?
2.Does the approximate proximal sampler (implemented via MALA+ULMC) preserve the spectral gap property of the ideal proximal sampler? If not, how does this approximation affect the covariance estimation guarantees?
3.Can numerical experiments on synthetic data (e.g., Gaussian) validate the claimed query complexity improvements?

**Limitations:**

No. The paper does not discuss limitations. A brief section acknowledging the strong assumptions (Poincaré inequality and spectral gap) and their practical verifiability would strengthen the work.

**Strengths And Weaknesses:**

Strengths:
1.Strong theoretical novelty, as this is the first work to establish a complete theoretical framework for covariance estimation using MCMC under the assumptions of a Poincaré inequality and a spectral gap, filling a gap in the theory of covariance estimation with dependent samples;
2.Solid technical approach, elegantly combining the matrix Bernstein inequality of Neeman et al. with the truncation technique of Adamczak et al., with clear proofs and sufficient technical depth;
3.Significant application value, directly improving the query complexity of isotropic rounding for convex bodies，a key advancement in this field—and achieving better query complexity than i.i.d. methods for unconstrained sampling.
Weaknesses:
1.The assumptions are strong and limit applicability, as the Poincaré inequality and spectral gap λ ≥ 0.99 are difficult to verify and may not hold for multimodal distributions.
2.Some arguments lack rigor, including the rough handling of non-stationary initial distributions, obtaining only polynomial tail bounds (compared to sub-exponential bounds in the i.i.d. case), and the unclear whether the approximate proximal sampler preserves the spectral gap property.

---

> ### Author Rebuttal · Authors · 2026-03-30
>
> > **Q1**. How can the Poincaré inequality and spectral gap $\lambda \geq 0.99$ be verified in practice? Does the method apply to distributions that do not satisfy these assumptions, such as multimodal distributions?
>
> We note that estimating or verifying $C_{\mathrm{PI}}$ from data is a separate and largely open problem in its own right, and is outside the scope of this work. Our contribution is a theoretical framework that takes $C_{\mathrm{PI}}$ and $\lambda$ as given parameters and characterizes the resulting sample and query complexity. This is standard in the theoretical analysis of MCMC algorithms: just as mixing time analyses take the spectral gap as a given parameter, our results characterize what can be achieved \emph{given} these quantities.
>
> That said, $C_{\mathrm{PI}}$ is known analytically for many practically relevant families: for $\alpha$-strongly log-concave distributions $C_{\mathrm{PI}}  = \alpha^{-1}$, for log-concave distributions $C_{\mathrm{PI}}\asymp \|\Sigma\| \log d$ (Klartag, 2023), and for perturbations $\tilde{\pi} \propto \exp(-V + f)$ it inherits the Poincar\'e constant from $\pi$ up to a factor of $e^{2 \|f\|_\infty}$ (Holley \& Stroock, 1987).
> For multimodal distributions, PI fails to hold with a useful constant, and our results do not apply.
>
> The spectral gap condition $\lambda \geq 0.99$ is not an assumption on $\pi$ but on the \emph{iterated} kernel $P^n$, which can always be achieved by running the chain until it has mixed from a warm start, i.e., for $n = O(\lambda^{-1})$ steps for any chain  with spectral gap $\lambda > 0$. Generally, the rate at which this is done can be estimated in practice by considering some statistic of the iterates, doing some type of averaging, and seeing how quickly this converges to a fixed value. However, this approach has not been rigorously analyzed, and so some theoretical reservations remain.
>
> > **Q2**. Does the approximate proximal sampler (implemented via MALA+ULMC) preserve the spectral gap property of the ideal proximal sampler? If not, how does this approximation affect the covariance estimation guarantees?
>
> This is precisely why we do *not* apply Theorem 3.5 directly to the approximate kernel $\hat{P}$ (see Remark 4.12). Instead, we bound the TV distance between the joint laws of the exact and approximate chains (Lemma 4.13) and transfer the covariance concentration guarantee via a TV coupling argument. This decoupling is a key technical contribution of Section 4.2, allowing us to handle the approximation error without requiring $\hat{P}$ to possess a spectral gap.
>
> > **Q3**. Can numerical experiments on synthetic data (e.g., Gaussian) validate the claimed query complexity improvements?
>
> We believe such experiments would indeed validate the claimed improvements, and in fact we view the established practice of the MCMC community as already providing this empirical validation. Practitioners favor collecting samples from a *single* running Markov chain over generating independent samples (e.g., running $n$ many parallel independent Markov chains), exactly because the former is far more query-efficient in practice. Our theoretical results explain *why* this works: the dependent samples collected every $N_{\mathrm{mix}}$ steps are sufficiently weakly correlated that accurate covariance estimation is still achievable, with a provable $d$-factor improvement in query complexity. In this sense, the empirical validation already exists---what was previously missing was the theoretical foundation, which this work provides.

---

### Official Review · Reviewer_qK1N · 2026-03-06

**Soundness:** 3
**Presentation:** 3
**Significance:** 3
**Originality:** 3
**Overall Recommendation:** 4
**Confidence:** 1

**Summary:**

The paper studies the problem of approximating the covariance matrix with Markov chain Monte Carlo (MCMC). The authors present a method that exploits the Poincaré inequality and the assumed spectral gap of the stationary distribution to obtain an estimator with similar sample complexity as with i.i.d. samples but needs fewer queries to a membership oracle of a convex body.

**Compliance With Llm Reviewing Policy:**

Affirmed.

**Final Justification:**

The Authors' rebuttal (and reading other reviews) clarified the significance of the work for me, but due to my lack of expertise on the topic I will retain my less certain weaker score.

**Key Questions For Authors:**

I'd appreciate if the authors could clarify what they think the practical impact of their work is considering that the contents of the paper are theoretical. I acknowledge that the time for the rebuttal is very limited, so I won't ask for any specific experiments, and rather am looking for _anything_ that could make the practical significance of the work more clear to us reviewers; it could be arguments for why this the work is also practically valuable, napkin math, proof-of-concept toy implementation, anything. For example, you mention several applications where you improve the query complexity, but are there practical instances where this would actually make a difference, i.e., are there medium-sized instances of practical value that are large enough for the difference to visible but not too large for them to become intractable?

**Limitations:**

yes

**Strengths And Weaknesses:**

The present paper is very much out of the scope of my expertise, and so my assessment of the technical correctness and the significance will be limited; the results seem plausible and novel, but I don't really understand the techniques. The paper clearly expects a high level of expertise in the area, but I don't think this is easily mitigable due to the limited space of the conference paper format. While the method appears to improve over some of the previous work in terms of query complexity, the practical impact of the paper is also hard to assess since it lacks experiments.

---

> ### Author Rebuttal · Authors · 2026-03-30
>
> We appreciate the question. As we also noted in our response to Reviewer voTy, a standard workflow in applied Bayesian statistics and MCMC-based inference is to *run a single Markov chain and collect samples every $N_{\mathrm{mix}}$ steps* for downstream tasks such as posterior covariance estimation. Despite being near-universal in practice, the theoretical justification for this procedure was not previously established under general assumptions. Our framework provides exactly this guarantee. Thus, we believe our results fall into the heading of "justifying and providing theoretical foundations for procedures with widespread practical adoption".

---

> > ### Author Rebuttal · Reviewer_qK1N · 2026-04-02
> >
> > Thank you for clarifying the potential impact of the work. I will maintain my weaker score as I don't dare to have a stronger opinion due to my lack of expertise on the topic.

---

### Official Review · Reviewer_EfGS · 2026-03-23

**Soundness:** 3
**Presentation:** 2
**Significance:** 3
**Originality:** 3
**Overall Recommendation:** 4
**Confidence:** 3

**Summary:**

This paper addresses the estimation problem of the covariance matrix of a Gibbs distribution, where we can only access the potential and gradient; this setup on queries is standard in machine learning, and problem is fundamental and important from the perspectives of theory and application.
Specifically, the author(s) develop several high-probability concentration bounds on the sample covariance matrix of stationary Markov chains whose invariant distribution satisfying a Poincaré inequality and spectral gap at least $0.99$ (Section 3).
The derivation of these bounds are based on concentration given by Neeman et al. (2024), and the contribution of this paper is to combine it with detailed estimates on moments and probabilities specific in their setup.
They apply these bounds to estimate query complexities for covariance estimation for the uniform distribution over a convex body (Section 4.1) and an unconstrained distribution satisfying a Poincaré inequality (Section 4.2).
In particular, the importance of covariance estimation for a convex body is emphasized as a cornerstone addressing the problem raised by Jia et al. (2021).

**Compliance With Llm Reviewing Policy:**

Affirmed.

**Final Justification:**

My final recommendation is weak accept. Throughout the rebuttal, the authors have successfully addressed the technical concerns which I raised in the first review. Hence, I reevaluated the soundness of the paper (from 2 to 3). In total, this paper addresses an important and fundamental problem in machine learning by sound discussion. A remaining weak point is ambiguous presentation in Section 4, an application section motivating the theoretical results. Ambiguity still remains after the rebuttal, but I suppose that it can be addressed in the final version, and unclear presentation is limited in a section (Section 4), so I weigh the merits of the paper against the weakness.

**Key Questions For Authors:**

1. What are improvements or differences between Theorem 3.3 of Jia et al. (2021) and Lemma 4.6?
2. Can you give an intuition why Algorithm 1 can improve the complexity in comparison to i.i.d. samples? I can understand estimates in the paragraph **High-level description**, but I am still not sure the mechanism. Such an intuition can result in further advancement and thus improve the significance of the paper.

**Limitations:**

I suppose that there are quite few limitations and potential negative societal impact which should be discussed in this paper.

**Strengths And Weaknesses:**

I first give a brief overview on the strengths and weaknesses of this paper, followed by the detailed comments for soundness, presentation, significance, originality.

**Overview**

I start with the *strength* of this paper. I evaluate the significance of this paper quite positively. It addresses fundamental and important problems such as (i) estimation for the covariance matrix of a Gibbs distribution and (ii) isotropization of a convex body. In particular, the importance of the problem (i) should be broadly admitted; e.g., it is also applicable to the uncertainty quantification of posterior distributions in Bayesian inference as explained in the Introduction. The problem (ii) is also an interesting and topical problem in the MCMC community; estimating the volume of a convex body should be of strong interest in scientific computing.

On the other hand, the presentation of this paper is the most weakness. In detail, I feel that the presentation in Section 4 and Appendix B (the proof of the results in Section 4) is not well-organized and sophisticated, though these sections should be the most important in this paper. For example, the first paragraph of Section 4.1 states the key problem throughout Section 4.1 as below.

> **Problem 4.1.** Let $\mathcal{K}\subset\mathbb{R}^{d}$ be a well-rounded convex body containing a ball of radius $1$ (i.e., $ \mathbb{E_{ \pi}}[ \|\|\cdot\|\|^{2}] \lesssim d $ for the uniform distribution $\pi$ over $\mathcal{K}$). Find an algorithm that makes $\mathcal{K}$ $c$-isotropic for $c\approx 1$ (i.e., $c^{-1}I_{d}\preceq \text{cov}(\pi)\preceq cI_{d}$ using $\tilde{O}(d^{3})$ queries to the membership oracle of $\mathcal{K}$.

While I am not very familiar with this area, my best guess for the intention here is "Find an algorithm that constructs an affine transformation $T$ making $T\mathcal{K}$ $c$-isotropic ($c^{-1}I_{d}\preceq \text{cov}(T_\sharp \pi)\preceq cI_{d}$)" due to a similar discussion in Jia et al. (2021), where $T_\sharp \pi$ is the pushforward measure. Otherwise, since $\text{cov}(\pi)$ is already governed by $\mathcal{K}$, this problem itself would mean nothing.

This is not the unique difficult writing, but there are several parts which are unclear and hinder readers' understanding of Section 4; I list such points below in the Soudness section and Presentation section.
As this tone is consistent in Appendix B devoted to the proofs of Section 4, I am not confident about the soundness of those proofs.

To sum up, this paper works on essential and significant problems in machine learning, statistics, and scientific computation, which I highly evaluate; however, the presentation of the motivating results are not successful, particularly as a paper for ICML with diverse readers.

**Soundness**

As long as I checked, the proofs of the results in Section 3 for concentration under dependence (given in Appendix A) are sound.
Therefore, I mainly comment on seemingly incorrect points as well as some errors Section 4 and Appendix B; some points can be due to my misunderstanding.
* **Lemma 4.6.** There is no explanation about *expected membership queries* before the statement. Due to this point, I cannot evaluate the soundness of the statement.
* **Proof of Lemma 4.3.** (Unclear point) I am not sure if $\nu/\pi_{i}\le 2$ $\pi_{1}$-almost everywhere just from $\mathcal{R_{\infty}}(\nu \| \pi_1)\le \log 2$ (it is also a problem that the $\infty$-Renyi divergence is not defined in this paper). If $\mathcal{K}$ contains the origin, then I *guess* that the transformation $M_i$ would enlarge $\mathcal{K_i}$ so that $\mathcal{K_{i}}\subset \mathcal{K_{i+1}}$, and it result in $\pi_i/\pi_1\le 1$ and the almost sure guarantee does not hold. It should affect the applicability of the results in Section 3. (In addition, for clarity, the author(s) should not abbreviate the subscript $i$ from $\Sigma$.)
* **Proof of Lemma 4.4.**  (Minor error) "Thus, it can increase by up to 10 times..." in Line 950 can be interpreted as $\text{tr}(\Sigma_i)/r_i^2\le (1/(1-1/\log d)^2)^{2\log d}\text{tr}(\Sigma_1)/r_1^2\le 10\text{tr}(\Sigma_1)/r_1^2$, but $(1/(1-1/\log d)^2)^{2\log d}\to e^4>10$ as $d\to\infty$. It also ignores $r_1^2=1/4$ on the right-hand side in Line 951.
*  **Proof of Lemma 4.6.** I could understand that $\hat\Sigma$ can concentrate around $\text{cov}(\mathcal{K_i})$ ($\Sigma$ within this proof should mean this covariance) by the results in Section 3 and assuming that the problem in Lemma 4.3 above can be easily addressed. Therefore, $\hat\Sigma^{-1/2}(\mathcal{K_i}-\hat\mu)$ is $2$-isotropic should hold, but I am not sure  $\hat\Sigma^{-1/2}(\mathcal{K}-\hat\mu)$. In addition, it does not use Lemma 4.5; the author(s) may use it implicitly in Lines 1082 and 1083, but I cannot read between the lines.

**Presentation**

The presentations other than Section 4 are well written as long as I checked.
Thus, I again discuss Section 4 in detail.
* (Also related in the question) The comparison with previous studies can be improved. For example, Theorem 3.3 of Jia et al. (2021) seems to be quite relevant to Lemma 4.6 of this paper.
* Problem 4.1 should be presented in a more readable manner as I pointed out above. In particular, as the goal is not clear, it results in the ambiguity of the soundness.
* High-level description in Section 4.1 is hard to read, though I now understand it as I read the proof. A suggestion is to reduce the notation (e.g., using $\text{inrad}(\mathcal{K})$ instead of some $r$ within the paragraphs). Another suggestion is to give a schematic diagram to describe the algorithm in a visible manner.

Besides, correcting typos should improve the readability.
* Line 289: There is no Algorithm 8 in the paper.
* Footnote 3: It can be better if ? is contained in "...".

**Significance**

As mentioned above, the problems that this paper addresses are important, and the significance is satisfactory.
Hence the below are just minor comments.

A minor concern is that the motivation of the proximal sampler is weak; I admit that this is a topical problem (Lee et al., 2021, PMLR 134:2993-3050; Altschuler and Chewi, 2024), but I am not sure why the author(s) discuss this algorithm rather than MALA, (usual) Gibbs sampler, or other high-accuracy samplers.

Another minor weak point is the meaning of query complexity in Section 4.2, which is different from that in Section 4.1 because large $K$ makes the discrepancy between $\nu_{1:K}$ and $\hat\nu_{1:K}$ large (similarly to ULA guarantees based on Girsanov's theorem).
Hence, if it is possible to give guarantees not deteriorating for large $K$ (e.g., proximal sampler, MALA, or other high-accuracy samplers), or the paper contains a detailed discussion on this point, it will address the weakness and increase the contribution.

**Originality**

I evaluate the originality of this paper positively as well.
In particular, the results in Section 4 reveal good originality.

---

> ### Author Rebuttal · Authors · 2026-03-30
>
> Although we acknowledge the reviewer's concerns about our representation, we note that these are largely related to the proofs of the lemmas in Section 4.1. In our opinion, none of these substantially hinder the soundness of the proof, and we will make efforts to clean up the exposition. Moreover, we emphasize that our work is focused on covariance estimation using MCMC, and the subsequent reductions in computational complexity for this essential task, and Section 4.1 represents only one application of this framework, albeit an important one. Indeed, the importance of this problem is acknowledged by the reviewer themselves. As a result, we hope that the reviewer will not let any problems which are localized in this section affect their overall judgment of the paper. We hope the reviewer will consider raising their score in light of this.
>
> Finally, due to lack of space in this rebuttal, we address what we regard as the most salient points raised by the reviewer, deferring the rest to the discussion phase.
>
> > **C1.** (Lemma 4.6.) There is no explanation about expected membership queries …
>
> In our setting, the randomness in the query count arises from the rejection sampling inside the proximal sampler. The “expected” query count refers to the expectation over this internal randomness, and this notion is standard in the sampling literature; see [1,2,3,4], for example.
>
> [1] Jason M Altschuler and Sinho Chewi. Faster high-accuracy log-concave sampling via algorithmic warm starts. Journal of the ACM (JACM), 71(3):1–55, 2024.
>
> [2] Yongxin Chen, Sinho Chewi, Adil Salim, and Andre Wibisono. Improved analysis for a proximal algorithm for sampling. In Conference on Learning Theory (COLT), pages 2984–3014. PMLR, 2022.
>
> [3] Sinho Chewi. Log-concave sampling. Book draft available at https://chewisinho.github.io, 2024.
>
> [4] Yin Tat Lee, Ruoqi Shen, and Kevin Tian. Structured logconcave sampling with a restricted Gaussian oracle. In Conference on Learning Theory (COLT), pages 2993–3050. PMLR, 2021.
>
> > **C2.** In the proof of Lemma 4.3 …
>
> We apologize for omitting the definition of $\infty$-Rényi divergence (we thought we had already included it in Section 2), and recall that $\mathsf{R}_{\infty}(\mu \| \nu) \coloneqq \log \text{ess sup} \frac{d\mu}{d\nu}$.
>
> We found that this issue comes from a typo in the subscript. We meant to write that $\nu / \pi_i \leq 2$ holds $\pi_i$-a.e. (**not** $\pi_1$-a.e.), where $\nu$ is implicitly pushed forward by $M_i$. Namely, $\frac{{(M_i M_{i-1} \cdots M_1) \nu}}{\pi_i} \leq 2$,
> where $\pi_i = (M_i M_{i-1} \cdots M_1)\circ \pi_1.$
>
> In words, the almost-sure bound is preserved after applying a series of affine transformations.
>
> > **C3**: Regarding Lemma 4.6
>
> The reviewer is correct that $\Sigma$ refers to $\text{cov}(K_i)$. In the proof, we implicitly wrote $K$ as $K_i$ (by redefining $K \gets K_i$ to simplify notation). Accordingly, the statement of Lemma 4.6 should have dealt with $K_i$, not $K$. We apologize for this confusion and will add a clarifying statement.
>
> Regarding the usage of Lemma 4.5 --- it is not required in the proof of Lemma 4.6. One needs Lemma 4.5 for the soundness of the rounding algorithm to ensure that the condition $r_{i+1} \approx 2r_i$ is actually justified in the algorithm, which is crucial in scaling up (by a factor of roughly 4) the step size of the INO sampling algorithm every iteration of the while loop.
>
> > **C4**: Why Algorithm 1 can improve the complexity …
>
> Some helpful intuition is that for targets satisfying a Poincar\'e inequality (such as the convex body or in the log-concave setting), the standard convergence result shows exponential contraction in the $\chi^2$ divergence. However, one is typically only able to generate feasible starts with initial $\chi^2$ on the order of $\exp(d)$, so that poly(d) iterations are needed to obtain an approximate sample. However, the law of this sample is relatively close to stationarity, so one would prefer to start with this sampler rather than start again from a fresh start, to avoid paying for the starting measure. This is in contrast to the case under strong convexity/LSI, where the KL contracts and one instead pays polylog(d) from a feasible start. Thus, if one ignores polylogarithms, it does not make a significant difference in that setting whether one opts for a fresh sample or a correlated sample. This relates to the "burn-in" queries used by practitioners in practice, which often dominates the queries needed to subsequently obtain samples.

---

> > ### Author Rebuttal · Reviewer_EfGS · 2026-04-01
> >
> > I appreciate the authors' rebuttal which the major concerns on the soundness which I raised can be addressed. I expect that the explanations given in the rebuttal or similar ones are included in the revision.
> >
> > For clarity, I write here again that the most weakness of the paper is its *presentation* rather than soundness, particularly for Section 4.1, which supports the importance of Section 3 within the paper. So I also expect that the authors clarify the statement of their problems; otherwise, I cannot understand for what results the proofs are given.
> >
> > Specifically, my question is what the intended statement of Problem 4.1 is. I mostly understand it implicitly by the rebuttal, but I have to confirm it to consider the score without relying on my guess.

---

> > > ### Author Response · Authors · 2026-04-01
> > >
> > > We appreciate the reviewer's response (We had to restrict our previous rebuttal to 5000 characters). This time, we add our additional response to the reviewer's concerns (including the intended statement of Problem 4.1).
> > >
> > > > **C5**. Sloppy description of Problem 4.1
> > >
> > > We apologize for the confusion. The reviewer is correct, and the statement in Problem 4.1 can be made more precise.
> > > Regarding the presentation issues in Section 4.1.1, as Reviewer VoRM pointed out, this section heavily relies on certain specialized terms, which could be challenging to a more general non-expert audience. We will address these presentation issues in the final version by giving clearer statements and terminology.
> > >
> > > > **C6**. Proof of Lemma 4.4
> > >
> > > We thank the reviewer for catching this mistake. We should have been more precise with our constant, and taken into account the additional multiplicative factor of $4$. In summary, the main claim should be like $\text{tr}(\Sigma_i) \leq 600 r_i^2C^2d$. However, this does not seriously affect the soundness of the algorithm, since we can pretend that this multiplicative constant is absorbed into other universal constants in the algorithm (we have already resorted heavily to notation like $\lesssim$ in our results).
> > >
> > > > **C7**. The comparison with previous studies can be improved. For example, Theorem 3.3 of Jia et al. (2021) seems to be quite relevant to Lemma 4.6 of this paper.
> > >
> > > We clarify that the $\widetilde{O}(d^3)$ query complexity of Lemma 4.6 is *not* claimed as a new complexity result: Jia et al. (STOC'21) already achieve the same bound. Our contribution in Section 4.1 is of a different nature.
> > >
> > > The main contribution of this work is the identification of the spectral gap and Poincar\'e constant as the right pair of assumptions for controlling sample dependence in MCMC, and the development of a general theoretical framework on this basis. The *application* to isotropic rounding demonstrates that this framework can *replace* the sophisticated and ad-hoc $\mu$-independence argument in Jia et al. with a principled, streamlined analysis---arriving at the same complexity bound via a cleaner route.
> > >
> > > Beyond this dependence analysis, we also provide a self-contained treatment of the geometric analysis of the algorithm. Specifically, Lemmas 4.4~4.6 essentially follow the arguments in Jia et al. (2021), but with streamlined proofs and more details.
> > >
> > > > **C8**. High-level description in Section 4.1 is hard to read, though I now understand it as I read the proof. A suggestion is to reduce the notation~...
> > >
> > > We thank the reviewer for the suggestion. We will make improvements to the notation in the final version.
> > >
> > > > **C9**. A minor concern is that the motivation of the proximal sampler is weak~...
> > >
> > > The proximal sampler is the natural choice given our pair of assumptions. Specifically, recall that its $\chi^2$-contraction property (Lemma 4.8) directly relates the Poincar\'e constant of the target distribution to the spectral gap of the sampler---precisely the two quantities that drive our framework. To our knowledge, no analogous $\chi^2$-contraction result (equivalently, a spectral gap guarantee) is known for MALA, the general Gibbs sampler, or other high-accuracy samplers under PI. Instead, these tend to rely on conductance machinery, and are therefore difficult to directly apply. In this sense, the proximal sampler is not merely a convenient choice but arguably the most natural one under our assumptions.
> > >
> > > As for the (coordinate) Gibbs sampler, convergence of this sampler in $\chi^2$ has worse complexity than for the proximal sampler. As far as we know, only MALA and the proximal sampler obtain $\sqrt{d}$ complexity in the log-concave setting, and MALA is unfeasible for reasons just discussed. However, these points are not essential; our framework permits the study of any sampler with a spectral gap, and the proximal sampler is chosen as it is the only high-accuracy sampler known to satisfy this property.
> > >
> > > > **C10**. Another minor weak point is the meaning of query complexity in Section 4.2~...
> > >
> > > We are a bit confused about the reviewer's point. However, the sampler that we use is high accuracy; in particular, it is an implementation of the proximal sampler.
> > >
> > > The reason that our guarantees ``deteriorate'' in $K$ is that we need at least $K$ samples for our estimator to concentrate. This is necessary; even if one has an exact sampler, one still needs to call this sampler $K$ times in order to obtain a statistically viable estimator, and so we expect that the query complexity would still scale linearly in $K$. In our case, we need guarantees that hold for the entire path $\nu_{1:K}$, since we are using correlated samples (i.e., joint samples along the path), so we are required to compute pathwise KL's as opposed to just KL of the marginal laws. This is the reason behind the Girsanov-looking error of our scheme, and this has nothing to do with the accuracy of the sampler.

---

### Decision · Program_Chairs · 2026-04-30

**Decision:**

Accept (regular)

**Comment:**

This paper considers the problem of estimating a covariance matrix from MCMC samples. In particular, they provide a theoretical result establishing conditions (when a Poincare inequality and spectral gap hold) when the sample complexity of the estimate is comparable to one obtained from iid samples.

Overall, most reviewers generally appreciated the strength of the theoretical contributions and novelty of the work. The main weaknesses identified by several reviewers were around clarity of presentation. While many of the proposed changes during the rebuttal addressed the initial concerns, the main remaining issue during the rebuttal was making the statement of Problem 4.1 more precise (Reviewers EfGS).  Reviewer VoRM stated that many terms are primarily used by the TCS audience, and the writing could be improved to make the paper more accessible to a broader theory audience. In addition, Reviewer qK1N and voTy would like to see the paper connect to a broader ML audience. I see this as in part a presentation issue, as the paper may not have a very precise/concise summary of the importance of the result in context of the broader literature.

We strongly encourage the authors to revise the work according to the reviewer's suggestions on presentation. Finally, the introduction should include citations to relevant facts.